# Canopy functional trait variation across Earth's tropical forests

Tropical forest canopies are the biosphere's most concentrated atmospheric interface for carbon, water and energy[1,2]. However, in most Earth System Models, the diverse and heterogeneous tropical forest biome is represented as a largely uniform ecosystem with either a singular or a small number of fixed canopy ecophysiological properties[3]. This situation arises, in part, from a lack of understanding about how and why the functional properties of tropical forest canopies vary geographically[4]. Here, by combining field-collected data from more than 1,800 vegetation plots and tree traits with satellite remote-sensing, terrain, climate and soil data, we predict variation across 13 morphological, structural and chemical functional traits of trees, and use this to compute and map the functional diversity of tropical forests. Our findings reveal that the tropical Americas, Africa and Asia tend to occupy different portions of the total functional trait space available across tropical forests. Tropical American forests are predicted to have 40% greater functional richness than tropical African and Asian forests. Meanwhile, African forests have the highest functional divergence—32% and 7% higher than that of tropical American and Asian forests, respectively. An uncertainty analysis highlights priority regions for further data collection, which would refine and improve these maps. Our predictions represent a ground-based and remotely enabled global analysis of how and why the functional traits of tropical forest canopies vary across space.

Tropical forests are the most biodiverse terrestrial ecosystems on Earth, and account for a large proportion of global diversity, including up to two-thirds of the approximately 73,000 tree species found on Earth[1]. They are responsible for key ecological functions, such as carbon exchange, nutrient cycling and the provision of water and energy[2], and they contribute to the livelihoods of more than a billion people around the world[5]. Despite the importance of canopy functional traits (morphological, physiological or phenological attributes that determine function) for forest responses to environmental change, our knowledge of the distribution of functional traits and of functional diversity at large spatial scales is limited, and this knowledge gap is particularly acute for tropical forests[6–8]. Although abiotic factors such as water availability, temperature and soil conditions are expected to drive variation in plant functional traits across spatial scales[9–11], we do not fully understand how these factors modulate canopy trait distributions and function[4]. Most global vegetation modelling efforts represent tropical forests as functionally uniform green slabs of canopy, incorporating little geographical variation in canopy functional properties[3]. This is due partially to the lack of spatially distributed functional trait data from across these regions[12]. In reality, the combination of climate, geology, evolutionary history and biogeography leads to complex but poorly understood trait variation[13]. There is, therefore, a fundamental need to describe and map how plant functional traits vary across tropical forests, because this variation has direct implications for ecosystem functioning and resilience to environmental change[14–16].

Predicting plant trait distributions across large spatial extents has generally focused on a few traits for which more observational data might be available, such as leaf nitrogen, leaf phosphorus and specific leaf area (SLA), and, in fewer cases, other leaf traits, such as leaf dry mass and leaf potassium[17–19]. Some advances in mapping trait distributions have been made by integrating plant functional type information with statistical modelling[17,19] and, more recently, satellite remote sensing[4,8]. However, most predictive models still make use of predefined plant functional types to estimate the distribution of single plant trait values, and still use coarse-resolution satellite data (for example, MODIS at 500 m) to map coarse indicators of community-level trait values—and often, few ground observations are available for tropical forests. This suggests the need to generate tools and methods that facilitate the tracking of functional traits across large spatial extents with high spatial and temporal resolution. Moreover, there is a need to develop methods to compare predictions of plant functional trait values created by different approaches[20]. Although plant trait databases[21,22] might help to model the distribution of functional traits as a function of biotic and abiotic conditions, we are far from having a full representation of the trait values for most tree species across the tropics, or even for single regions, such as Amazonia, with around 15,000 tree species[23]. Understanding functional trait variability across continents is crucial for predicting ecosystem responses to environmental change, including climate change and land-use alterations[9]. Previous work[24] revealed substantial variation in functional traits across different ecosystems, both within and between plant communities. This variation highlights the relationship between plant trait strategies and environmental conditions, which allows species to occupy distinct ecological niches.

## Tree traits across the tropics

Here we present the distribution of plant traits across the entirety of the planet's tropical forests by expanding on a methodology[6] that uses an approach to predict functional traits using the European Space Agency's Sentinel-2 satellite data. We used data for 13 tree functional

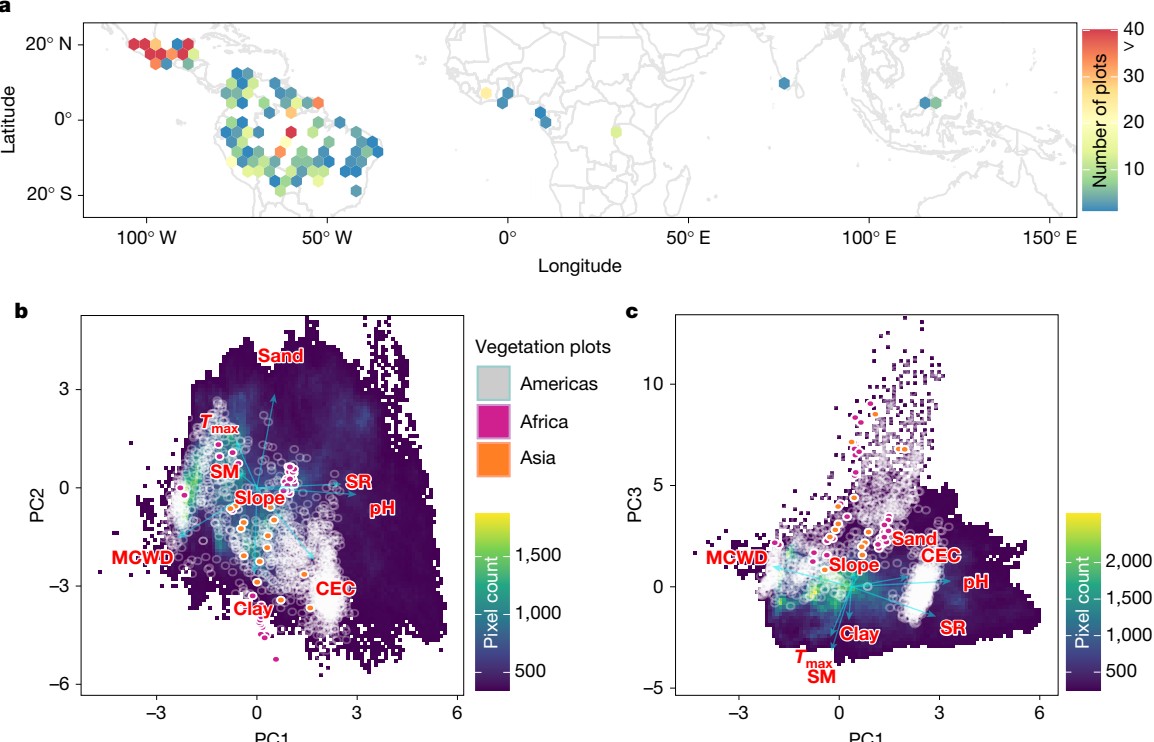

**Fig. 1 | Study area and PCA. a**, Study area, showing the distribution of 1,814 vegetation plots across the original biome space for tropical forests (grey background) in the Americas (659.6 ha), Africa (124.6 ha) and Asia (15.4 ha). **b,c**, PCA (PC1 and PC2, **b**; PC3, **c**) depicting the environmental space found across the tropics (yellow and green colours show higher map pixel counts representing area covered) on the basis of mean maximum air temperature ($T_{max}$), soil moisture (SM), solar radiation (SR), slope, MCWD, soil cation-exchange capacity (CEC), soil pH, sand amount and clay amount. The grey, violet and

orange points show the location of the sampling plots in environmental space found across the tropics. PC1 accounts for 27% of explained variance, PC2 for 24% and PC3 for 14%, with all three accounting for 65% of the total explained variance. PC1 is loaded mainly by water deficit index (MCWD) (−0.47), SR (0.50) and soil pH (0.59); PC2 by the soil sand (0.57), clay (−0.53) and CEC (−0.44); and PC3 by SM (−0.63) and $T_{max}$ (−0.49). Climate data were derived for each pixel from the TerraClimate project[34] and soil data were derived from SoilGrids.org.

traits (hereafter referred to as plant traits), spanning leaf morphological (leaf area, SLA, thickness, fresh and dry mass, also including leaf water content) and chemical (mass-based calcium, carbon, magnesium, nitrogen, potassium and phosphorus concentrations) traits, and also including predictions for wood density[24,25]. These plant traits were gathered from across tropical forests from the Americas, Africa and Asia, here including northeast Australia in our broad definition of Asian tropical forests (Fig. 1a). We focus on upper-canopy leaf traits, which are the main interface for forest–atmosphere exchange (in that they are part of key processes such as transpiration and photosynthesis[26]) and which are directly detectable by spectral remote sensing. The plant traits are hence related to fundamental aspects of leaf morphology, chemistry and tree structure (Extended Data Table 1).

Overall, we expect that acquisitive traits, which enhance the efficient capture and use of resources (for example, high SLA and leaf nutrient content), will be more prominent in locations with pronounced seasonal variation and nutrient-rich soils. By contrast, conservative traits (for example, thicker, nutrient-poor leaves, high wood density) are likely to dominate in areas with less seasonal variability and poorer soils. In forests dominated by deciduous species, such as drier tropical forests, we expect species with acquisitive traits to become more prevalent, thereby making these traits more common in the ecosystem. African forests, which have experienced a long-term drying trend[27], generally exhibit lower species diversity[28] and distinct soil conditions[29] compared with American and Asian tropical forests. We expect these differences to result in a narrower distribution of plant trait values when compared with the wetter tropical forests of the Americas and Asia. In addition, Asian tropical forests contain the widespread distribution of the *Dipterocarpaceae* family[30], which we anticipate will mainly define

the particular set of dominant traits in those areas, such as those associated with large, tough leaves, which are characteristic of this tree family.

Traits were collected from the Global Ecosystems Monitoring (GEM) network[31], ForestPlots.net[32], BIEN (https://bien.nceas.ucsb.edu/bien/), TRY (www.try-db.org) and a previous study[33]. We incorporated vegetation census data from the GEM and Monitoreo Nacional Forestal (MONAFOR) networks and contributing networks to ForestPlots.net, with geolocated tree individuals from 1,814 permanent vegetation plots (Fig. 1a), spanning a wide set of environmental conditions across tropical forests (Fig. 1b) and covering a total of 799.5 ha (Extended Data Table 2). We used the plant traits and vegetation censuses to create pixel-level (from the Sentinel-2 satellites) community weighted mean (CWM) trait values using a previously described method[6]. The total number of CWM pixels used in our analysis was 79,955, which were distributed across 18 countries in the 4 tropical continents (Extended Data Table 2). Our vegetation plots are more abundant in the tropical forests of the Americas and could be thought to represent the environmental conditions in this region rather than those in Africa and Asia. Our principal component analysis (PCA) (Fig. 1b,c) shows that although our sampling sites do not cover all environmental space available across the tropics—especially those climates that are less common in the tropics (dark purple zone in Fig. 1b,c)—we fundamentally cover the most prominent environmental conditions found across tropical forests.

For each pixel for which we calculated trait CWM, we also extracted surface reflectance data from the Sentinel-2 satellite bands covering the years 2019–2022. On the basis of these spectral bands, we also generated the modified chlorophyll absorption reflectance index (MCARI), modified soil adjusted vegetation index 2 (MSAVI2) and normalized difference red edge index (NDRE). Using the grey-level co-occurrence

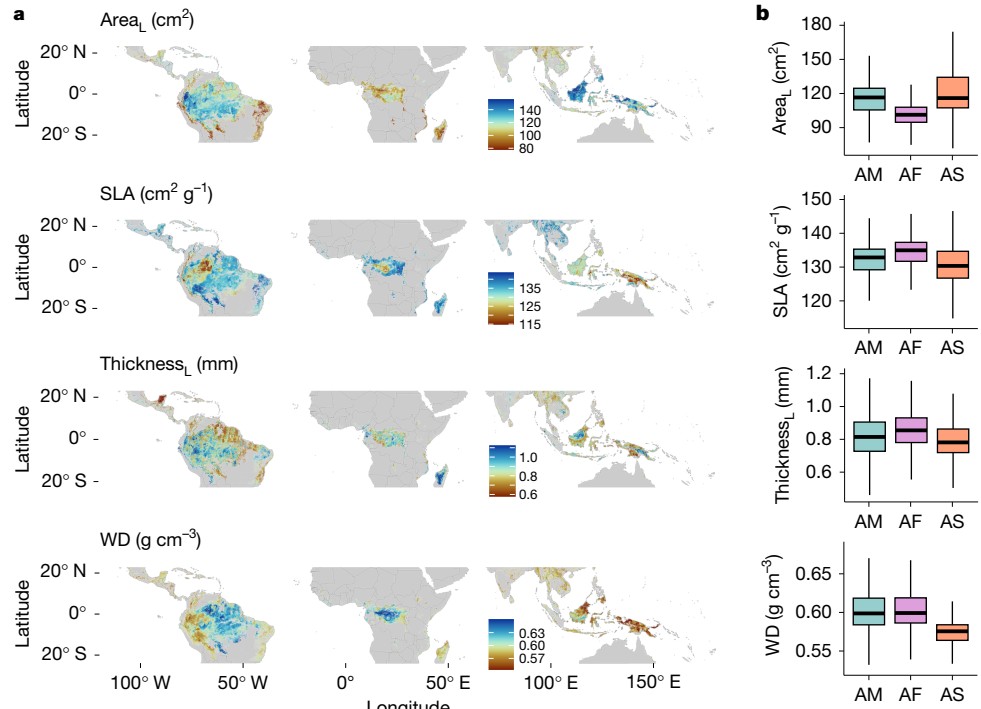

**Fig. 2 | Predicted distribution of CWM morphological and structural plant traits. a**, Predicted distribution of a selection of CWM morphological and structural plant traits. Red to orange show areas with low to intermediate trait values; light to dark blue depict areas with intermediate to high trait values. The remaining morphological traits and the spatial predictions of their uncertainty are shown in Supplementary Figs. 1–7. **b**, Box plots showing the CWM trait distribution values for tropical American (AM), African (AF) and Asian (AS) forests, extracted from the spatial predictions. The horizontal black lines depict the median CWM trait value and vertical lines show the whiskers extending to the largest CWM trait value or not further than 1.5 times the interquartile range. For visualization purposes, we excluded the extreme lowest and highest 1% of values in the maps in **a** and outliers in **b**. Area$_L$, leaf area; Thickness$_L$, leaf thickness; WD, wood density. For statistical model results, see Supplementary Table 1. For the significance of differences between CWM trait mean values, obtained using a *t*-test with Bonferroni correction, see Supplementary Table 2.

matrix (GLCM) for these indices, we calculated their entropy and correlation as canopy texture variables. We extracted soil texture and chemistry (clay percentage, sand percentage, pH and cation-exchange capacity (CEC)) across the sampling plots from SoilGrids.org and joined these with terrain (slope) and climate (maximum climatic water deficit (MCWD) and maximum temperature ($T_{max}$)) from the TerraClimate dataset[34]. We used the above-mentioned covariates in random forest models that have produced accurate plant-trait-mapping results[4,6] to predict CWM plant functional traits at a pantropical scale. Our analysis hence provides insights into the variation in plant trait composition across climatic and soil gradients across tropical forests. We tested for the prediction accuracy and uncertainty in trait predictions while accounting for potential spatial autocorrelation using a plot-level spatial block leave-one-out cross-validation[35] (Supplementary Table 1). We present the spatial predictions from the statistical models described above for canopy-level morphological traits, wood density (Fig. 2 and Supplementary Figs. 1–7) and chemistry (Fig. 3 and Supplementary Figs. 8–13). Using our 13 plant trait model predictions (maps), we tested fundamental knowledge gaps on the functional composition across tropical American, African and Asian forests.

Models for leaf chemistry and wood density showed higher accuracy (mean $R^2$ = 0.66 and 0.48, respectively) than did those for leaf morphology traits (mean $R^2$ = 0.25; Supplementary Table 1). Among these, leaf nitrogen (mean $R^2$ = 0.53, root mean squared error (RMSE) = 0.29), phosphorus (0.50, 0.02) and calcium (0.64, 0.22) concentrations had the highest prediction accuracy, followed by leaf carbon (0.40, 1.42) and potassium (0.46, 0.17). Models for SLA (0.32, 19.95), leaf dry mass (0.32, 0.58) and leaf fresh mass (0.31, 2.24) showed moderate accuracy scores. By contrast, leaf magnesium concentration (0.27, 0.06), leaf area (0.22, 66.15), leaf water content (0.18, 3.92), and leaf thickness

0.17, 0.79) had lower accuracy. As expected, lower explanatory values were found when testing the models with the plots from Africa or Asia separately, because fewer data were available (Supplementary Table 1). The individual surface reflectance of the Sentinel-2 bands, the derived vegetation indices and the climate and terrain variables obtained on average the highest importance scores across traits, with texture and soil metrics obtaining on average lower importance values (Extended Data Fig. 1). We report variable importance scores per variable and plant trait in Supplementary Figs. 1f–13f.

We make available our trait mapped predictions across the tropics as an online resource in which more detail can be obtained across the tropical region (https://pantropicalanalysis.users.earthengine.app/view/pantropical-traits-aguirre-gutierrez-2025). Using the modelled trait maps (Figs. 2a and 3a and Supplementary Figs. 1–13), we compared the CWM trait values among continents, which provided insights into the variations in plant traits across continents (Supplementary Table 2, Figs. 2b and 3b and Supplementary Figs. 1–13). Following our predictions, for most traits, Asian forests show some of the highest average canopy-level trait values; specifically, average leaf area (119.3 cm$^2$), leaf calcium (0.88%), potassium (0.79%) and magnesium (0.28%) concentrations, leaf water content (54.8%), leaf fresh (3.9 g) and dry (1.06 g) mass. These findings are supported by local plot-level data[6]. However, similar values were found for leaf phosphorus for Asia and Africa (0.11%) and slightly lower for the Americas (0.10%), and also for leaf carbon (around 47%) and leaf nitrogen concentrations (around 2.15%). African forests are predicted to have, on average, the smallest leaves (average of 100 cm$^2$), highest leaf thickness (0.85 mm) and SLA (133.9 cm$^2$ g$^{-1}$). Wood density is predicted to be, on average, higher in tropical American and African forests (around 0.60 g cm$^{-3}$), as suggested previously[36]. These results emphasize Asia's unique trait spectra

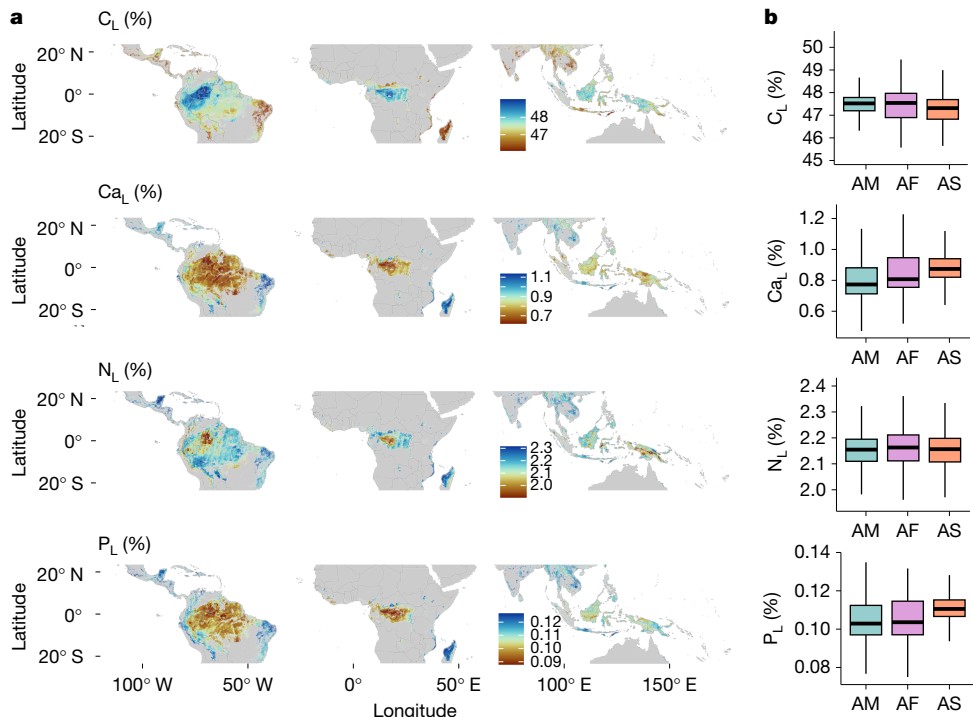

**Fig. 3 | Predicted distribution of CWM leaf nutrient plant traits. a**, Predicted distribution of a selection of CWM leaf nutrient plant traits. Red to orange show areas with low to intermediate trait values; light to dark blue depict areas with intermediate to high trait values. The remaining chemistry traits and the spatial predictions of their uncertainty are shown in Supplementary Figs. 8–13. **b**, Box plots showing the CWM trait distribution values for tropical American (AM), African (AF) and Asian (AS) forests, extracted from the spatial predictions. The horizontal black lines depict the median CWM trait value and vertical lines show the whiskers extending to the largest CWM trait value or not further than 1.5 times the interquartile range. For visualization purposes, we excluded the extreme lowest and highest 1% of values in the maps in **a** and outliers in **b**. $C_L$, leaf carbon concentration; $Ca_L$, leaf calcium concentration; $N_L$, leaf nitrogen concentration; $P_L$, leaf phosphorus concentration. For statistical model results, see Supplementary Table 1. For the significance of differences between CWM trait mean values, obtained using a *t*-test with Bonferroni correction, see Supplementary Table 2.

## Traits in wet and dry tropical forests

A changing climate affects the distribution and persistence of forests across the tropics. There is an ongoing debate about the capacity of wet and dry tropical forests to adapt or shift their functional composition given global environmental change[38]. Studies have shown that drier tropical forests could be responding faster to a changing climate by shifting their trait composition[39], but also that such drier tropical forests might be becoming more functionally homogeneous, which could negatively affect their capacity to respond to further environmental change[37]. Hence, understanding the distribution of key tree functional traits across tropical forests is crucial to understanding their potential response to environmental change, including climate.

We determined the extent of tropical broadleaf wet and dry forests using the RESOLVE Ecoregions dataset[40]. On the basis of this division, wet forests, on average, had higher leaf area and leaf carbon concentration than did dry forests (Supplementary Fig. 1c and Supplementary Table 3). By contrast, dry forests, which are characterized by the presence of stronger and longer dry seasons, had higher average values for leaf chemistry traits such as leaf magnesium, nitrogen, calcium, potassium and phosphorus, and also for SLA (Supplementary Figs. 1d–13d and Supplementary Table 3). These distinct strategies possibly ensure optimal nutrient use for drought avoidance, on the basis of the leaf economics spectrum of 'low' leaf construction costs for fast energy gains amid challenging environmental conditions[41]. Notably, both wet and dry tropical forests converge in certain traits, with comparable mean levels of leaf thickness, dry and fresh mass, leaf water content

and wood density underscoring their shared strategies. However, these similar average trait values could also be due to the fact that both strategies—drought avoidance and drought tolerance—can be present across both wet and dry forests, potentially ensuring resilience across tropical forest types[42]. These findings from our comprehensive trait predictions provide crucial insights into the intricate linkages between environmental factors and plant traits across continents, contributing to our understanding of ecological diversity and adaptation strategies in diverse tropical forest ecosystems. Our findings shed light on the diverse plant trait patterns observed across continents, enhancing our understanding of global ecological variation[24].

Areas across the wet tropics, which are highly species-diverse, tended to have slightly more uncertain predictions (that is, higher standard error; s.e.) for most traits than did drier tropical forests (Supplementary Figs. 1–13, middle panel). Our results for leaf morphology and tree structural traits such as fresh mass and wood density showed higher uncertainty in predictions (s.e. = 0.4–1.6 g and 0.02–0.05 g cm$^{-3}$ respectively) across wetter locations such as central Amazonia, central Africa and Borneo. However, for most other morphological and leaf nutrient traits, their prediction uncertainty was low in most of the tropics (Supplementary Figs. 1–13, middle panel). Overall, the uncertainty in the predictions of some traits might result from searching for simple relationships between individual traits and the environment, whereas tree individuals represent a combination of traits and trait values that might be interpreted as functional strategies or syndromes. It is the syndrome rather than the individual trait that is selected for in nature. Our findings on the uncertainty of trait predictions give an insight into areas across the tropics that could benefit the most from more extensive field trait campaigns (Supplementary Figs. 1–13, middle panel and Extended Data Fig. 2).

and how the African flora is adapted to a wide range of current and past environmental conditions[37].

## Functional diversity of tropical forests

The resilience of an ecosystem to environmental change can be partially assessed by the diversity of its functional trait values. According to the insurance hypothesis about biodiversity and ecosystem functioning[43], ecosystems with greater taxonomic and functional diversity could potentially be less affected by changes in the environment. Recent studies support this, showing that tropical forests with higher functional diversity and high functional redundancy tend to be less adversely affected by extreme weather events such as El Niño than do less functionally diverse and redundant forests[44]. Hence, functional diversity indicators such as functional richness and functional divergence can shed light on the capacity of ecosystems to respond to global environmental change. Determining the functional diversity of tropical forest ecosystems will therefore enhance our understanding of their resilience and the possible effects of environmental change on ecosystem functioning and its services to people.

To generate a pantropical understanding of the functional diversity of tropical forests across the Americas, Africa and Asia, and to ascertain how these three regions compare, we first built a PCA that offers insights into the distribution of ecological strategies or syndromes of plant communities[45] across tropical forests. This PCA was based on the pixel values from the spatial predictions (maps) of canopy and wood density traits (Figs. 2 and 3 and Supplementary Figs. 1–13). The first two PCA axes (Fig. 4a,b), explain 44% (PC1) and 20.6% (PC2) of the pantropical trait variance, respectively, and highlight key traits that drive the functional space across tropical forests at a pantropical extent. In our analysis, leaf nutrients such as calcium, nitrogen, phosphorus, potassium and magnesium are the main traits loading PC1 (−0.39, −0.25, −0.39, −0.39 and −0.38, respectively; Supplementary Table 4), with carbon (0.35) and wood density (0.27) in opposite directions. PC2 is loaded mainly by leaf structural and morphological characteristics such as dry mass (0.52), fresh mass (0.43), area (0.47) and SLA (−0.32) (Fig. 4a,b).

Following the PCA results, central-west Amazonia, central Africa and to some extent some areas of Southeast Asia show areas with trait syndromes related to higher wood density and leaf carbon (Fig. 4c, top, PC1), but also higher leaf area and leaf fresh and dry mass (Fig. 4c, middle, PC2). Wood density is closely related to plant mechanical and hydraulic properties, and has been shown to have a negative relationship with mortality given increased physical strength and resistance to drought-induced embolism[46]. The highest leaf carbon concentration (C) values are predicted to be found in wet regions with relatively infertile soils in the Americas, Africa and Asia, such as northwest Amazonia, Central Africa and much of Borneo, and tend to decline towards drier tropical forests (Fig. 3a and Supplementary Fig. 8). An alternative strategy for dry forest tree species is deciduousness, which leads to low leaf carbon concentration because of lower investment in leaf defence and longevity. In dry forests with fertile soils, we expect deciduousness as a dominant strategy (thus low C), but in less fertile soils we would expect a transition to an evergreen strategy (higher C) to conserve resources. Higher leaf carbon, and generally also higher leaf fresh and dry mass, reflect an increased investment in leaf structural and physical defences[47], which favours longer leaf life span and thus higher investment in compounds such as lignin, tannins and soluble phenolics that contain high levels of carbon[48].

Syndromes related to higher leaf nutrients (Fig. 4c, top, PC1) and higher SLA (Fig. 4c, middle, PC2) are opposed to the patterns explained above, with higher leaf nutrients and intermediate SLA values found across tropical dry forests and increasing leaf water content predicted across the Andes and high elevations of Southeast Asia (Fig. 4c, bottom, PC3). Leaf nutrients are generally lowest in wet central-west Amazon, Central Africa and wet forests of insular Southeast Asia (Fig. 3), and tend to increase across dry forests in south and southeastern Brazil, West Africa, eastern Madagascar and most of the tropical forests in India and northern Southeast Asia (Figs. 3 and 4, PC3). This suggests that soil physical and chemical properties have an important role in shaping leaf phosphorus distributions[49] (Figs. 3a, bottom and 4a,c and Supplementary Fig. 13). We predict a consistently high leaf area across much of insular Southeast Asia (Fig. 4c). This agrees with previous plot-level analyses[31] that found a larger leaf area for forests in Malaysian Borneo than for those from other tropical regions. Many of the wet Bornean forest canopies are dominated by a single family (*Dipterocarpaceae*)[30] with a particular set of traits, such as large, tough leaves, and this biogeographical feature might explain some of the leaf morphological differences between Asian and other forests. In the tropical Americas, syndromes related to lower SLA values are found across the Andes, the mountains of southern Brazil and also in the extremely wet and nutrient-poor areas of northwest Amazonia; for example, across the sandy soils of upper Rio Negro. Lower SLA can be found across Central Africa and in Asia across the mountains of New Guinea (Fig. 4c, bottom). Plants with a lower SLA tend to have thicker leaves, which are more resistant to herbivory and decomposition, and lower SLA values indicate a conservative strategy in which resources are invested in long-lasting leaves but often with a lower photosynthetic capacity[25].

Building on our PCA analysis, we calculated the trait functional diversity, here by means of the trait functional richness (FRich) and functional divergence (FDiv), across tropical forests (Fig. 4a), and calculated how these FRich and FDiv values differ between the forests of the tropical Americas, Africa and Asia (Fig. 4b). FRich represents the size of the functional trait space and FDiv indicates the distribution of CWM trait abundances within the functional trait space[45]. The overall FRich across tropical forests is calculated to be 111.7, with a pantropical FDiv of 0.46 (Fig. 4a). The observed FRich values of 109.2 for the Americas, 66.5 for Africa and 63.5 for Asia point to large differences in the diversity of functional trait values in these regions (Fig. 4b). The higher FRich of the Americas suggests that these forests have a broader array of plant strategies and adaptations, potentially influenced by diverse environmental niches and historical factors[44], and congruent with the fact that the most taxonomically diverse tropical forests are in the tropical Americas[1,23]. By contrast, the lower FRich in Africa and Asia suggests that specific environmental filters or historical biogeographical constraints shape the functional traits of plant communities in these regions[50]. The FDiv values ranged from 0.42 for the Americas to 0.61 for Africa, and 0.57 for Asia, revealing varying degrees of dissimilarity in functional trait space among tropical forests (Fig. 4b). Higher FDiv values imply greater divergence, suggesting stronger niche differentiation or competitive interactions. The comparatively lower FDiv in the Americas might imply a higher degree of functional redundancy across communities. Conversely, the higher FDiv in Africa and Asia suggests a more specialized pattern of resource use, owing potentially to intense interspecific competition or specific ecological constraints in these regions. Regions with higher functional divergence might exhibit higher ecosystem stability because of niche complementarity, whereas regions with lower divergence might face challenges in adapting to changing environmental conditions. The observed patterns have implications for ecosystem functioning, biodiversity conservation and ecosystem services provision.

Understanding the tree trait composition and functional diversity across the tropics is of pivotal importance for global biodiversity and ecosystem modelling and for conservation efforts[51]. Although dynamic global vegetation models (DGVMs) and species distribution models (SDMs) help to assess the effects of a changing climate, DGVMs often rely on broad plant functional types and SDMs commonly overlook functional trait composition and diversity (but see ref. 52). By incorporating trait-based mechanisms and functional trait diversity, models can better capture the variability in plant responses, potentially making more realistic predictions related to carbon cycling[53], vegetation distribution[54] and ecosystem composition and resilience[44]. DGVMs and

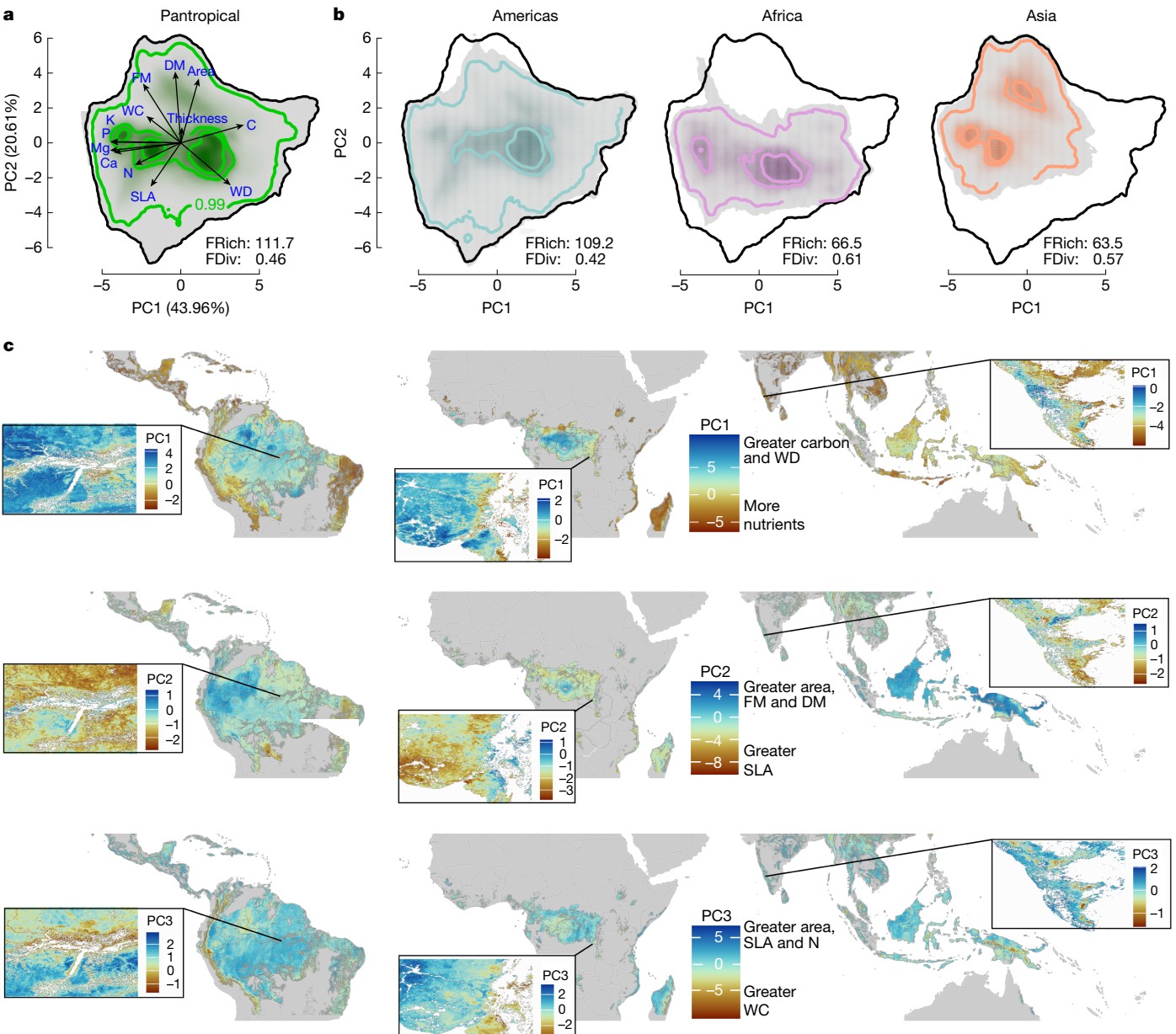

**Fig. 4 | Functional diversity of tropical forests in the Americas, Africa and Asia. a**, Functional trait space of trees across tropical forests in the Americas, Africa and Asia (including Australia), with principal component PC1 explaining 44% and PC2 20.6% of the variance in plant traits distributions. Arrows indicate the contribution and direction of each trait for the PCA. **b**, Distribution of functional trait space for the tropical American (left), African (middle) and Asian (right; including Australia) forests separately. **a** and **b** show the probabilistic density distribution defined by the PC1 and PC2 space of the 13 plant functional traits used: area, leaf area; C, leaf carbon concentration; Ca, leaf calcium concentration; K, leaf potassium concentration; Mg, leaf magnesium concentration; N, leaf nitrogen concentration; P, leaf phosphorus concentration;

DM, leaf dry mass; FM, leaf fresh mass; SLA, specific leaf area; thickness, leaf thickness; WC, leaf water content; WD, wood density (see Extended Data Table 1 for a description of the trait used). The inner colour gradient represents the density of pixels in the PC trait space. Thick contour lines depict the 0.5 and 0.99 quantiles. FRich shows the functional richness and FDiv the functional divergence for the global trait space across continents (**a**) and for tropical American (**b**, left), African (**b**, middle) and Asian (**b**, right) forests. **c**, PC1 (top), PC2 (middle) and PC3 (bottom, explaining 13% of the variance) from **a** predicted across tropical forests. Co-occurring trait syndromes or strategies are shown, with insets magnified to show greater details of the predicted plant strategies.

SDMs could include plant traits and plant functional diversity estimates to advance our understanding of ecosystem functioning and responses to global environmental change.

Our capacity to use artificial intelligence (AI) to map plant functional traits by means of deep-learning models applied to field trait[55] data or even photographs[56] is quickly developing. These models can process vast amounts of remote-sensing data to identify and classify diverse biodiversity metrics[57]. Some models—particularly convolutional neural networks—have been integrated with spectral data to map plant

traits using field data[58] and also, recently, citizen-science approaches[56]. New satellites with hyperspectral capabilities and high spatial resolution are in development, and the availability of tree censuses and trait data across the tropics is increasing. This will open new avenues for testing the capabilities of large machine-learning models, possibly involving deep learning, for using data across time and space from multiple sources. However, to obtain robust and reliable indicators of plant functional diversity and biodiversity levels across ecosystems, AI models should complement and not replace conventional ecological

methods—especially the direct field sampling and botanical identification of individual trees by experts. There is a need for tools that can generate predictions of biodiversity at high temporal resolution, and our approach represents a way forward in this direction. Looking ahead, there is the potential to track plant functional diversity across time (for example, on a yearly basis) using satellite remote-sensing data, such as that from the Sentinel-2 satellites. Such an application would require major efforts in terms of field ecological data collection, availability of new satellite data, modelling algorithms, computing power and storage capabilities. All of this can be achieved by strong and fair collaborations between field researchers, universities and other public and private research organizations.

Our study reveals and maps the geographical variation in the functional composition of the tropical moist and dry forests, where at least two-thirds of Earth's tree species are found[1]. Our trait predictions indicate deep physiological constraints of adaptation to long-term climate; the predictions could provide the basis for forecasting how shifting climates will affect the functional composition of tropical forests, and could help to develop a more mechanistic understanding and realistic predictive ecology across spatio-temporal scales. Built from unique, geolocated field records combined with an array of spectral, textural and environmental data, our maps represent data-informed spatial hypotheses that will assist in the identification of priority areas for further field data collection, especially across tropical forests in Africa and Asia, where fewer data are available. The ultimate accuracy of the plant functional trait predictions depends on the sample coverage, the accuracy of the field measurements and the quality of the pantropical covariates that are used to spatially extrapolate our models. Undoubtedly, predictions will improve as new environmental datasets become available and as vegetation census and trait data expand further over space and time. Nevertheless, these maps represent a major improvement on previous site-based speculation for analysing geographical variation in the ecophysiology of the entire tropical forest biome, and they thereby inform our understanding of how tropical forests function in the context of the whole Earth system.

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

Jesús Aguirre-Gutiérrez[1,2]✉, Sami W. Rifai[3], Xiongjie Deng[1], Hans ter Steege[4,5], Eleanor Thomson[1], Jose Javier Corral-Rivas[6], Aretha Franklin Guimaraes[7], Sandra Muller[8], Joice Klipel[8,105], Sophie Fauset[9], Angelica F. Resende[10,11], Göran Wallin[1,12], Carlos A. Joly[13,14], Katharine Abernethy[11,15], Stephen Adu-Bredu[16,17], Celice Alexandre Silva[18], Edmar Almeida de Oliveira[19], Danilo R. A. Almeida[10], Esteban Alvarez-Davila[20], Gregory P. Asner[21], Timothy R. Baker[22], Maíra Benchimol[23], Lisa Patrick Bentley[24], Erika Berenguer[1,25], Lilian Blanc[26], Damien Bonal[27], Kauane Bordin[28], Robson Borges de Lima[29], Sabine Both[30], Jaime Cabezas Duarte[31,32], Domingos Cardoso[33,34], Haroldo C. de Lima[34], Larissa Cavalheiro[36], Lucas A. Cernusak[36], Nayane Cristina C. dos Santos Prestes[19], Antonio Carlos da Silva Zanzini[37], Ricardo José da Silva[18], Robson dos Santos Alves da Silva[18], Mariana de Andrade Iguatemy[34,38], Tony César De Sousa Oliveira[39,40], Benjamin Dechant[41,42], Géraldine Derroire[26,43], Kyle G. Dexter[44,45,46], Domingos J. Rodrigues[35], Mário Espírito-Santo[47], Letícia Fernandes Silva[48,49], Tomas Ferreira Domingues[50,51], Joice Ferreira[52], Marcelo Fragomeni Simon[53], Cécile A. J. Girardin[1], Bruno Hérault[26], Kathryn J. Jeffery[11], Sreejith Kalpuzha Ashtamoorthy[54], Arunkumar Kavidapadinjattathil Sivadasan[54], Bente Klitgaard[55], William F. Laurance[36], Maurício Lima Dan[56], William E. Magnusson[7], Eduardo Malta Campos-Filho[57], Rubens Manoel dos Santos[58], Angelo Gilberto Manzatto[59], Marcos Silveira[60], Ben Hur Marimon-Junior[61], Roberta E. Martin[21], Daniel Luis Mascia Vieira[53], Thiago Metzker[62,63], William Milliken[64], Peter Moonlight[65], Marina Maria Moraes de Seixas[52], Paulo S. Morandi[66], Robert Muscarella[67], María Guadalupe Nava-Miranda[68,69], Brigitte Nyirambangutse[70,71], Jhonathan Oliveira Silva[72], Imma Oliveras Menor[1,73], Pablo José Francisco Pena Rodrigues[34], Cinthia Pereira de Oliveira[29], Lucas Pereira Zanzini[74], Carlos A. Peres[75], Vignesh Punjayil[54], Carlos A. Quesada[76], Maxime Réjou-Méchain[73], Terhi Riutta[1,77], Gonzalo Rivas-Torres[78], Clarissa Rosa[7], Norma Salinas[79], Rodrigo Scarton Bergamin[80,81], Beatriz Schwantes Marimon[61], Alexander Shenkin[82], Priscyla Maria Silva Rodrigues[72], Axa Emanuelle Simões Figueiredo[83], Queila Souza Garcia[84], Tereza Spósito[62], Danielle Storck-Tonon[85], Martin J. P. Sullivan[86], Martin Svátek[87], Wagner Tadeu Vieira Santiago[88], Yit Arn Teh[89], Prasad Theruvil Parambil Sivan[54], Marcelo Trindade Nascimento[90], Elmar Veenendaal[91], Irie Casimir Zo-Bi[92], Marie Ruth Dago[92], Soulemane Traoré[92,93], Marco Patacca[94], Vincyane Badouard[43,73,92], Samuel de Padua Chaves e Carvalho[95], Lee J. T. White[11,15], Huanyuan Zhang-Zheng[1,2], Etienne Zibera[12,96], Joeri Alexander Zwerts[97], David F. R. P. Burslem[98], Miles Silman[99,100], Jérôme Chave[101], Brian J. Enquist[102,103], Jos Barlow[25], Oliver L. Phillips[22], David A. Coomes[104] & Yadvinder Malhi[1,2]

[1]Environmental Change Institute, School of Geography and the Environment, University of Oxford, Oxford, UK. [2]Leverhulme Centre for Nature Recovery, University of Oxford, Oxford, UK. [3]School of Biological Sciences, University of Adelaide, Adelaide, South Australia, Australia. [4]Naturalis Biodiversity Center, Leiden, The Netherlands. [5]Quantitative Biodiversity Dynamics, Utrecht University, Utrecht, The Netherlands. [6]Facultad de Ciencias Forestales y Ambientales, Universidad Juárez del Estado de Durango, Durango, Mexico. [7]Coordenação de Biodiversidade, Instituto Nacional de Pesquisas da Amazônia, Manaus, Brazil. [8]Plant Ecology Lab, Ecology Department, Universidade Federal do Rio Grande do Sul, Porto Alegre, Brazil. [9]School of Geography, Earth and Environmental Sciences, University of Plymouth, Plymouth, UK. [10]Department of Forest Sciences, Luiz de Queiroz College of Agriculture, University of São Paulo (USP/ESALQ), Piracicaba, Brazil. [11]Biological and Environmental Sciences, University of Stirling, Stirling, UK. [12]Department of Biological and Environmental Sciences, University of Gothenburg, Gothenburg, Sweden. [13]Departamento de Biologia Vegetal, Instituto de Biologia, Universidade Estadual de Campinas, Campinas, Brazil. [14]Brazilian Platform on Biodiversity and Ecosystem Services (BPBES), Campinas, Brazil. [15]Institut de Recherche en Écologie Tropicale, Libreville, Gabon. [16]CSIR–Forestry Research Institute of Ghana, Kumasi, Ghana. [17]Department of Natural Resources Management, CSIR College of Science and Technology, Kumasi, Ghana. [18]Universidade do Estado de Mato Grosso, Tangará da Serra, Brazil. [19]PPG Ecology and Conservation, Universidade do Estado de Mato Grosso, Nova Xavantina, Brazil. [20]Escuela de Ciencias Agrícolas, Pecuarias y Ambientales – ECAPMA, Universidad Nacional Abierta y a Distancia, Bogota, Colombia. [21]Center for Global Discovery and Conservation Science, Arizona State University, Tempe, AZ, USA. [22]School of Geography, University of Leeds, Leeds, UK. [23]Laboratório de Ecologia Aplicada à Conservação, Departamento de Ciências Biológicas, Universidade Estadual de Santa Cruz, Ilhéus, Brazil. [24]Department of Biology, Sonoma State University, Rohnert Park, CA, USA. [25]Lancaster Environment Centre, Lancaster University, Lancaster, UK. [26]Forêts et Sociétés, Université de Montpellier, CIRAD, Montpellier, France.

[27]INRAE, Université de Lorraine, AgroParisTech, UMR Silva, Nancy, France. [28]Department of Ecology, Universidade Federal do Rio Grande do Sul, Porto Alegre, Brazil. [29]Laboratório de Manejo Florestal, Universidade do Estado do Amapá, Macapá, Brazil. [30]Environmental and Rural Science, University of New England, Armidale, New South Wales, Australia. [31]Jardín Botánico de Bogotá, Bogotá, Colombia. [32]Universidad de los Andes, Bogotá, Colombia. [33]Instituto de Biologia, Universidade Federal da Bahia, Salvador, Brazil. [34]Instituto de Pesquisas Jardim Botânico do Rio de Janeiro, Rio de Janeiro, Brazil. [35]Universidade Federal de Mato Grosso, Sinop, Brazil. [36]College of Science and Engineering, James Cook University, Cairns, Queensland, Australia. [37]Departamento de Engenharia Florestal, Universidade Federal de Lavras, Lavras, Brazil. [38]Instituto Internacional para Sustentabilidade, Rio de Janeiro, Brazil. [39]Institute of Biogeosciences, IBG2: Plant Sciences, Forschungszentrum Jülich GmbH, Jülich, Germany. [40]Faculty of Communication and Environment, Hochschule Rhein-Waal, Kamp-Lintfort, Germany. [41]German Centre for Integrative Biodiversity Research (iDiv) Halle-Jena-Leipzig, Leipzig, Germany. [42]Leipzig University, Leipzig, Germany. [43]Cirad, UMR EcoFoG (AgroParistech, CNRS, INRAE, Université des Antilles, Université de la Guyane), Campus Agronomique, Kourou, French Guiana. [44]Royal Botanic Garden Edinburgh, Edinburgh, UK. [45]School of GeoSciences, University of Edinburgh, Edinburgh, UK. [46]Department of Life Sciences and Systems Biology, University of Turin, Turin, Italy. [47]Departamento de Biologia Geral, Universidade Estadual de Montes Claros, Montes Claros, Brazil. [48]Universidade Paulista, Polo Rio Branco, Brazil. [49]Universidade Federal do Acre, Rio Branco, Brazil. [50]Faculdade de Filosofia, Ciências e Letras de Ribeirão Preto, Ribeirão Preto, Brazil. [51]Universidade de São Paulo, São Paulo, Brazil. [52]Embrapa Amazônia Oriental, Belém, Brazil. [53]Embrapa Recursos Genéticos e Biotecnologia, Brasília, Brazil. [54]Forest Ecology Department, KSCSTE–Kerala Forest Research Institute, Kerala, India. [55]Department for Accelerated Taxonomy, Royal Botanic Gardens, Kew, Richmond, UK. [56]Centro de Pesquisa, Desenvolvimento e Inovação Sul, Instituto Capixaba de Pesquisa, Assistência Técnica e Extensão Rural, Cachoeiro de Itapemirim, Brazil. [57]Instituto Socioambiental, São Paulo, Brazil. [58]Laboratório de Fitogeografia e Ecologia Evolutiva, Departamento de Ciências Florestais, Universidade Federal de Lavras, Lavras, Brazil. [59]Departamento de Biologia, Universidade Federal de Rondônia, Porto Velho, Brazil. [60]Centro de Ciências Biológicas e da Natureza, Universidade Federal do Acre, Rio Branco, Brazil. [61]Laboratório de Ecologia Vegetal (LABEV), Universidade do Estado de Mato Grosso, Nova Xavantina, Brazil. [62]IBAM—Instituto Bem Ambiental, Belo Horizonte, Brazil. [63]Myr Projetos Sustentáveis, Belo Horizonte, Brazil. [64]Royal Botanic Gardens, Kew, Richmond, UK. [65]Botany, School of Natural Sciences, Trinity College Dublin, Dublin, Ireland. [66]Programa de Pós-graduação em Ecologia e Conservação, Universidade do Estado de Mato Grosso, Nova Xavantina, Brazil. [67]Plant Ecology and Evolution, Department of Ecology and Genetics, Uppsala University, Uppsala, Sweden. [68]Colegio de Ciencias y Humanidades, Universidad Juárez del Estado de Durango, Durango, Mexico. [69]Escuela Politécnica Superior de Ingeniería, Universidad de Santiago de Compostela, Campus Terra, Lugo, España. [70]Global Green Growth Institute, Rwanda Program, Kigali, Rwanda. [71]University of Rwanda, Kigali, Rwanda. [72]Colegiado de Ecologia, Universidade Federal do Vale do São Francisco (UNIVASF), Senhor do Bonfim, Brazil. [73]AMAP, Université de Montpellier, IRD, CNRS, CIRAD INRAE, Montpellier, France. [74]Departamento de Engenharia Florestal, Universidade do Estado de Mato Grosso, Caceres, Brazil. [75]School of Environmental Sciences, University of East Anglia, Norwich, UK. [76]Coordenação de Dinâmica Ambiental, Instituto Nacional de Pesquisas da Amazônia, Manaus, Brazil. [77]Department of Geography, University of Exeter, Exeter, UK. [78]Estación de Biodiversidad Tiputini, Colegio de Ciencias Biológicas y Ambientales, Universidad San Francisco de Quito, Quito, Ecuador. [79]Institute for Nature Earth and Energy, Pontificia Universidad Católica del Perú, Lima, Peru. [80]School of Geography, Earth and Environmental Sciences, University of Birmingham, Birmingham, UK. [81]Birmingham Institute of Forest Research (BIFoR), University of Birmingham, Birmingham, UK. [82]School of Informatics, Computing, and Cyber Systems, Northern Arizona University, Flagstaff, AZ, USA. [83]Departamento de Engenharia Florestal, Universidade de Brasília, Brasília, Brazil. [84]Instituto de Ciências Biológicas, Departamento de Botânica, Universidade Federal de Minas Gerais, Belo Horizonte, Brazil. [85]Programa de Pós-graduação em Ambiente e Sistemas de Produção Agrícola, Universidade do Estado de Mato Grosso, Tangará da Serra, Brazil. [86]Department of Natural Sciences, Manchester Metropolitan University, Manchester, UK. [87]Department of Forest Botany, Dendrology and Geobiocoenology, Faculty of Forestry and Wood Technology, Mendel University in Brno, Brno, Czech Republic. [88]CESAM—Centro de Estudos do Ambiente e do Mar, Departamento de Biologia, Pesquisador Colaborador, Universidade de Aveiro, Aveiro, Portugal. [89]School of Natural and Environmental Sciences, Newcastle University, Newcastle upon Tyne, UK. [90]Laboratório de Ciências Ambientais, CBB, Universidade Estadual do Norte Fluminense Darcy Ribeiro, Campos dos Goytacazes, Brazil. [91]Plant Ecology and Nature Conservation Group, Wageningen University and Research, Wageningen, The Netherlands. [92]UMRI SAPT (Sciences Agronomiques et Procédés de Transformation), Institut National Polytechnique Félix Houphouët-Boigny, Yamoussoukro, Côte d'Ivoire. [93]Ministry of Water and Forests, Abidjan, Côte d'Ivoire. [94]Forest Ecology and Forest Management Group, Wageningen University and Research, Wageningen, The Netherlands. [95]Federal Rural University of Rio de Janeiro, Seropedica, Brazil. [96]School of Forestry and Biodiversity and Biological Sciences, College of Agriculture, Animal Sciences and Veterinary Medicine, University of Rwanda, Musanze, Rwanda. [97]Utrecht University, Utrecht, The Netherlands. [98]School of Biological Sciences, University of Aberdeen, Aberdeen, UK. [99]Center for Energy, Environment, and Sustainability, Wake Forest University, Winston-Salem, NC, USA. [100]Department of Biology, Wake Forest University, Winston-Salem, NC, USA. [101]Centre de Recherche Biodiversité Environnement, CNRS, UPS, IRD, Université de Toulouse, INPT, Toulouse, France. [102]Department of Ecology and Evolutionary Biology, University of Arizona, Tucson, AZ, USA. [103]The Santa Fe Institute, Santa Fe, USA. [104]Conservation Research Institute and Department of Plant Sciences, University of Cambridge, Cambridge, UK. [105]Present address: Institute of Ecology, Leuphana University of Lüneburg, Lüneburg, Germany. ✉e-mail: jeaggu@gmail.com

## Methods

### Vegetation plots and plant traits

We gathered vegetation census data from the GEM and MONAFOR networks and contributing networks to ForestPlots.net, being geolocated tree individuals from 1,814 demarcated and identified vegetation plots (Fig. 1a). The vegetation plots covered a wide set of the environmental conditions found across tropical forests (Fig. 1b) and spanned 799.5 ha (Extended Data Table 2). We aimed to match each individual tree to a trait value. All plant functional traits used are part of the Global Ecosystems Monitoring network (GEM; gem.tropicalforests.ox.ac.uk)[31], the MONAFOR network, the ForestPlots (www.ForestPlots.net)[32,59,60], BIEN (https://bien.nceas.ucsb.edu/bien/) and TRY (www.try-db.org)[22] databases and from local collaborators and Diaz et al.[33], and were collected following a standardized methodology described in Both et al.[61], Martin et al.[62], Enquist et al.[63], Oliveras et al.[50] and Gvozdevaite[64]. For the traits provided by the GEM network and ForestPlots.net, the tree species that contributed the most to plot basal area were sampled with three to five replicate individuals per species. Species representing 80% or more of the basal area were sampled for traits in low-diversity sites and at least 70% in high-diversity sites. For each selected tree, a sun and a shade branch were sampled, and in each branch, three to five leaves were used for trait measurements. Leaf samples were analysed for chemistry (nitrogen, phosphorus, carbon, calcium, potassium and magnesium concentration) and morphological and structural traits (area: area, specific leaf area (SLA); thickness: thickness; fresh mass (FM); and water content (WC); see Extended Data Table 1 for units and definitions). If more than one value per trait per species was available, we used the trait mean at the species level for subsequent analysis. Our approach aimed to cover at least 70% of the canopy area of a pixel within a plot with trait data at species or genus level, often covering more than that (Extended Data Fig. 3). Because when species-level trait data were unavailable we used the mean genus-level data, our analysis could be seen as more representative of the genus-level trait responses. When achieving at least 70% coverage was not possible for a given trait in a given pixel, that pixel was left out of the analysis for the specific trait. All species names were standardized following the Taxonomic Name Resolution Service (TNRS; https://tnrs.biendata.org).

### Calculating community-level trait values

We used the pixel-level CWM trait method from Aguirre-Gutiérrez et al.[6] in our analysis, in which they calculated the CWM of each trait for each 10 × 10-m pixel of the Sentinel-2 imagery on the basis of the canopy area occupied by the single tree crowns of each species encompassed in a given pixel. The total number of CWM pixels used in our analysis was 79,955, from 1,814 unique permanent forest plots distributed across 18 countries in the 4 tropical continents (Extended Data Table 2). A full detailed description of the methods can be found in Aguirre-Gutiérrez et al.[6], and we summarize it here. We calculated the CWM trait values for each 10 × 10-m Sentinel-2 pixel falling into a vegetation plot. We first geolocated the vegetation plot and the distribution of each individual tree in the plot. Some of the plots already had their tree crowns mapped. When this was not the case, we calculated the crown area using regional allometric equations, from which we generated a crown polygon. Then, for each pixel we calculated the trait CWM using the individual tree crown horizontal area as the weighting factor. We used only pixels that had at least a 70% basal area coverage with trait value to generate the trait CWM.

### Sentinel-2 spectral data

The European Space Agency Sentinel-2 satellites (sentinel.esa.int/web/sentinel/missions/sentinel-2) have high multispectral (13 spectral channels covering the visible, near-infrared and short-wave infrared), spatial (10 m for visible and near-infrared 835 nm, 20 m for other near-infrared and short-wave infrared) and temporal coverage (revisit period of

5 days), in addition to open data availability. This high spatial, radiometric and temporal resolution provides the backbone to scale functional traits, such as leaf morphology, water content and covalent chemical bonds, without the logistical and field constraints that are common across the tropics[6] and other regions[65]. The extraction of Sentinel-2 Level-2A data on surface reflectance bands, vegetation indices and canopy texture metrics has been fully described previously[6], and here we give a summary of the main steps. We extracted Sentinel-2 Level-2A spectral data at the pixel level for each vegetation plot using the raw band values for bands B2 to B12, excluding bands B9 and B10 because those are used for cirrus, water vapour and cloud detection for the images and dates specified in Supplementary Table 5. Next, we calculated the vegetation indices MCARI, MSAVI2 and NDRE.

We also incorporated spatial information by using the spectral indices to derive neighbourhood canopy texture, entropy and correlation with a 9 × 9-pixel GLCM (ref. 66). The GLCM metrics are computed from a matrix that is spatially dependent. The co-occurrence matrix relies on the angular orientation and distance between adjacent pixels, illustrating the frequency of associations between a pixel and its neighbouring pixels. We applied a 9 × 9-pixel kernel window because this window size proved sufficient to capture ample canopy contrast information during the modelling stage without incurring substantial computation time.

We generated spatially explicit predictions across tropical forests in Google Earth Engine (GEE)[67] using surface reflectance Sentinel-2 Level-2A images from June to March of 2019–2022, because these months show the lowest cloud cover across most of our study areas. We applied the maskS2clouds and maskEdges to increase the quality of the imagery, especially to detect and mask clouds and cirrus. On the basis of the images selected, we calculated a median spectral reflectance composite value per band and used it for generating the predictive maps. The reader can run the GEE code (Supplementary Table 5) to obtain the number and identity of the imagery used.

### Climate, topography and soil data

We used the TerraClimate climate dataset[34] to extract climate data for the study area. These data have an original spatial resolution of around 4.6 km at the Equator and a large temporal range (from 1951 to the present). In general, the TerraClimate dataset builds on the Climatic Research Unit climate data, CRU (refs. 54,68), downscales it and swaps the JRA55 reanalysis product[69] for CRU when there is insufficient station data to inform CRU. From the TerraClimate dataset, we calculated the 30-year (encompassing 1988–2017) mean annual $T_{max}$ and the MCWD for each vegetation plot. The MCWD is a metric for drought intensity and severity defined as the most negative value of the climatological water deficit (CWD) of a given year, and we calculated it following a previous study[70] but using the potential evapotranspiration instead of a fixed evapotranspiration value. We derived topography (slope) from the Shuttle Radar Topography Mission (SRTM) digital elevation model V3 product (SRTM Plus) provided by NASA JPL at an original spatial resolution of around 30 m at the Equator[71]. Soil characteristics such as texture and fertility also determine the distribution of plant species[47]. Moreover, drier tropical forests tend to be distributed on more nutrient-rich soils than do wetter forests[72], which would therefore also select for species adapted to such conditions. Maps of soil data—that is, per cent sand and clay, pH and CEC—were obtained from the SoilGrids project (https://soilgrids.org)[73] at a spatial resolution of 250 m per pixel. All climate, topography and soil datasets were scaled to the Sentinel-2 pixel resolution to take advantage of its spectral reflectance pixel size. All spatial analyses were performed in the GEE platform.

### Mapping plant traits

We modelled each plant functional trait CWM as a function of the spectral, soil, topography and climatic variables using the random forests (RF) machine-learning algorithm[74] in the R platform[75] with the Ranger function in a high-performance computing system. RF stands out as a

nonparametric algorithm known for its capabilities against overfitting and for its flexibility with respect to variations in the type and number of variable inputs. This robustness is attributed to the bagging process and the inclusion of random feature selection. In addition, RF has been widely and successfully applied for modelling and predicting ecological and remote-sensing data, both within individual ecosystems and across diverse environments[6,65,76–78]. To parametrize the RF models, we performed a comprehensive series of model optimization and regularization techniques to mitigate overfitting[6]. We determined the number of trees through a cross-validation analysis, exploring a range between 500 and 1,500 trees. Similarly, we varied the number of variables randomly sampled as candidates at each split (also referred to as 'mtry' in the RF) in the range of 1–10. The final model incorporated the combination of parameters that yielded the lowest RMSE. We then obtained a map by applying the fitted model to make predictions for the full tropics, where tropical wet and tropical dry broadleaf forests are located (because the data used for model fitting belong to these forest types). We determined the extent of the tropical broadleaf wet and dry forest using the RESOLVE Ecoregions dataset[40] (https://ecoregions.appspot.com/) and the tropical countries boundaries dataset (for the GEE app)[79]. We further used the Land-use Cover map from the European Space Agency[80] to delimit the areas classified as forest and a previously described[81] 30-m forest cover product to further delimit the predictions to areas with a threshold value of a minimum of 25% forest cover in a given pixel. Hence, although an area might be included in the trait maps, this does not mean that it is entirely forested. The accuracy of the predictions was quantified by the explained variance using $R^2$. Variable importance was calculated as the decrease in node impurities, from splitting on the focus variable, derived from the out-of-bag error. We scaled the variable importance values per covariate to a 0–1 scale for comparison purposes.

To assess the uncertainty in model predictions in a spatially explicit manner, we used spatial leave-one-out cross-validation[35] for the full dataset. When predicting the RF models, we also obtained their s.e. using the infinitesimal jackknife approach as a measure of prediction uncertainty. From these s.e.-mapped predictions, we also calculated a final map of new field sampling needs by standardizing each trait s.e.-mapped prediction from 0 to 1 and obtaining an average value of the sum of those standardized s.e. maps. From this final field sampling needs map, we calculated the areas belonging to the lowest, middle and highest 33 percentiles and classified these as low, intermediate and high, respectively. This final map could aid in generating field sampling priorities for the traits used in this study.

We tested for differences in the among-continent mean CWM trait values using $t$-test analysis with Bonferroni correction for significance values. Because we are working with the pixel predictions per continent (here using 100 × 100 m pixels), we have several millions of pixel-level estimates, which makes it possible to obtain significant $P$ values ($P < 0.05$) just because of the high number of pixels involved. Therefore, we performed the $t$-test for the full dataset (comparing continents) and also by first randomly sampling 10% and 1% of the data per continent for the comparisons so as to obtain an indication of the possible effect of sample size on the among-continent comparison results.

## Functional richness and divergence

We calculated the FRich and FDiv at a pantropical extent and also for the tropical Americas, Africa and Asia. To this end, we took the mapped CWM trait predictions and performed a PCA with them and calculated the trait probability density (TPD) as described before[45,82] Using the mapped predictions, and not only the pixels used to build the trait CWM, allowed us to avoid having a larger representation of the tropical forests in the Americas in comparison to those found in Africa and Asia. To perform the PCA, we used the Princomp function in R with the data from the mapped predictions of the 13 traits. We then used the Funspace function in R to create the TPDs, with which we would obtain

the functional trait space available at a pantropical extent. We also calculated the TPDs for each continent on the basis of the pantropical TPD so that these could be compared between each other[45,82]. On the basis of these, we then calculated the FRich and FDiv metrics at a pantropical extent and also for each continent. In our analysis we represent the global TPD (100%) and also highlight the contours containing 50% and 99% of the total probability.

## Reporting summary

Further information on research design is available in the Nature Portfolio Reporting Summary linked to this article.

## Data availability

To comply with the original data owners' requirements, the plant functional traits and vegetation census data that support the findings of this study are available from their sources: GEM[31] at gem.tropical-forests.ox.ac.uk, ForestPlots[32,59,60] (www.ForestPlots.net) and Diaz et al.[33] Because of the data sovereignty from the original data owners, raw data on vegetation censuses and trait data are not publicly available, but can be requested by contacting all researchers through the ForestPlots[32,59,60] data request protocol described at https://forest-plots.net/en/join-forestplots/working-with-data. The processed maps with community-level trait predictions from this study are available as an app in GEE at https://pantropicalanalysis.users.earthengine.app/view/pantropical-traits-aguirre-gutierrez-2025. Other environmental and plant data are available from their original sources: BIEN (https://bien.nceas.ucsb.edu/bien), SoilGrids (https://soilgrids.org) and RESOLVE Ecoregions (https://ecoregions.appspot.com). Satellite data from Sentinel-2 are freely available from the GEE platform (https://developers.google.com/earth-engine/datasets/catalog/COPERNICUS_S2_SR_HARMONIZED).

## Code availability

R code for graphics and analyses is available on Zenodo at https://doi.org/10.5281/zenodo.14509493 (ref. 83).

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

**Acknowledgements** J.A.-G. was funded by the Natural Environment Research Council (NERC) under an Independent Research Fellowship (NE/T011084/1), the NERC Pushing the Frontiers (NE/Z504191/1) and the Oxford University John Fell Fund (10667). This paper is a product of several vegetation data networks including the GEM network (gem.tropicalforests.ox.ac.uk), RAINFOR, MONAFOR and the ForestPlots.net meta-network. This manuscript is an output of ForestPlots.net research projects 109 and 184, 'Predicting plant functional traits across the tropics'. ForestPlots.net is a meta-network and cyber-initiative developed at the University of Leeds that unites permanent plot records and supports tropical forest scientists. We acknowledge the contributions of the ForestPlots.net Collaboration and Data Request Committee (T.R.B., E. Honorio Coronado, A. Levesley, O.L.P., B.S.M., B. Sonké, C. Ewango, J. Muledi, S. Lewis, L. Qie) for facilitating this project and associated data management. The development of ForestPlots.net and data curation has been funded by multiple grants including NE/B503384/1, NE/N012542/1 – 'BIO-RED', ERC Advanced Grant 291585 – 'T-FORCES', NE/F005806/1 – 'AMAZONICA', NE/N004655/1 – 'TREMOR', NERC New Investigators Awards, the Gordon and Betty Moore Foundation ('RAINFOR', 'MonANPeru'), ERC Starter Grant 758873 – 'TreeMort', and from EU Framework 5, 6 and 7. Global trait collection and trait analyses for GEM were funded by an ERC Advanced Investigator Award (GEM-TRAIT: 321131) to Y.M. under the European Union's Seventh Framework Programme (FP7/2007–2013), with additional support from NERC grant NE/D01174/1 and NE/J022616/1 for trait work in Peru, NERC grant ECOFOR (NE/K016385/1) for trait work in Santarem, NERC grant BALI (NE/K016369/1) for plot and trait work in Malaysia and ERC advanced grant T-FORCES (291585) to O.L.P. and Y.M. for trait work in Australia. Plot set-up in Ghana and Gabon was funded by NERC grant NE/I014705/1 and by the Royal Society–Leverhulme Africa Capacity Building Programme. The Malaysia campaign was also funded by NERC grant NE/K016253/1. Plot inventories in Peru were supported by funding from the US National Science Foundation (NSF) Long-Term Research in Environmental Biology program (LTREB; DEB 1754647) and the Gordon and Betty Moore Foundation Andes–Amazon Program. Plot inventories in Nova Xavantina (Brazil) were supported by the National Council for Scientific and Technological Development (CNPq), Long Term Ecological Research Program (PELD), process 441244/2016-5 and the Foundation of Research Support of Mato Grosso (FAPEMAT), Project ReFlor, process 589267/2016. The network of long-term permanent plots in Mexico, MONAFOR, is supported by the National Forest Council (CONAFOR), National Council of Humanities Science and Technology (CONAHCYT) and Council of Science and Technology of the State of Durango (COCYTED). Trait data acquisition in Gabon was supported by the Gabon National Parks Agency. H.Z.-Z. was supported by NERC NE/T011084/1 grant to J.A.-G. and by NGEE–Tropics, funded by the U.S. Department of Energy, Office of Science, Office of Biological and Environmental Research. S.A.-B. acknowledges funding from the Leverhulme Trust—Royal Society of the United Kingdom (A130026) under the water stress, ecosystem function and tree functional diversity in tropical African forests project. C.A.J. acknowledges support from the Brazilian National Research Council–CNPq (PELD process 403710/2012–0), NERC and the State of São Paulo Research Foundation (FAPESP) as part of the projects Functional Gradient, PELD/BIOTA and ECOFOR (processes 2003/12595-7, 2012/51509-8 and 2012/51872-5, within the BIOTA/FAPESP Program–the Biodiversity Virtual Institute (www.biota.org.br); COTEC/IF 002.766/2013 and 010.631/2013 permits. B.S.M. was supported by CNPq–PELD projects (441244/2016-5 and 441572/2020-0) and CAPES (136277/2017-0). M.S. acknowledges funding for the Andes Biodiversity and Ecosystem Research Group (ABERG) plot network from the US NSF LTREB (1754647), the Gordon and Betty Moore Foundation's Andes to Amazon Initiative and RAINFOR. E.B., J.B. and Y.M. acknowledge support from NERC under projects NE/K016431/1 and NE/S01084X/1. Y.M. is supported by the Frank Jackson Foundation. A.F.R. acknowledges support from FAPESP 22/14605-0 and 19/24049-5; G.W. from the Swedish Research Council grant 2021-05265 and the Swedish strategic research area 'Biodiversity and Ecosystem services in a Changing Climate' (BECC; http://www.becc.lu.se/); E.A.d.O. from CNPq, Brazil 153023/2022-8 and 150666/2023-3; D.R.A.A. from FEALQ; K.B. from the Instituto Nacional de Ciência e Tecnologia (INCT) in Ecology, Evolution and Conservation of Biodiversity, from MCTIC–CNPq 465610/2014-5 and from FAPEG grant 201810267000023; R.B.d.L. from UEAP; J.C.D. from COL-TREE; L.C. from UFMT/Sinop; D.J.R. from CNPq (productivity grant -312407/2022-0), T.F.D. from CNPq Bolsa Produtividade 312589/2022-0; B.K. from the Kerala Forest Research Institute Plan Fund; B.K. from a Darwin Initiative main grant 20-021; W.E.M. from CNPq (productivity grant), FAPEAM and CNPq (funding for PPBio, PELD, CENBAM and smaller projects); C.R. from INCT-SinBiAm (grant CNPq 406767/2022-0), INCT-CENBAM (grant CNPq 406474/2022-2), CAPES/FAPEAM (grant 88887.964874/2024-00) and PPBio (grant CNPq 441260/2023-3 and 441228/2023-2); E.M.C.-F. from the Good Energies Foundation; R.M.d.S. from CNPq Bolsa de Produtividade 313632/2021-9; A.G.M. from FAPERO/PAP/Universal/AP-CA/2022, CNPq (funding for PPBio, PELD, CENBAM and smaller projects); W.M. from the Darwin Initiative main grant 20-021; R.M. from the Swedish Research Council (Vetenskapsrådet), grant 2019-03758; C.P.d.O. from UEAP; M. Svátek from a grant from the Ministry of Education, Youth and Sports of the Czech Republic (INTER-TRANSFER LTT19018); and M.T.N. from CNPq (Research Productivity Fellowship) grant 312567/2021-9 and FAPERJ (E-26/201.007/2022). B.D. was supported by sDiv, the Synthesis Centre of iDiv (DFG FZT 118, 202548816). D.C.'s research in plant biodiversity is supported by grants from CNPq (Research Productivity Fellowship no. 314187/2021-9 and PPBio Semiárido no. 441271/2023-5) and FAPERJ (Programa Jovem Cientista do Nosso Estado - 2022, grant no. 200.153/2023). S.W.R. acknowledges support from the Australian Research Council Discovery Grant (DP190101823) and from the ARC Centre of Excellence for Climate Extremes (CE170100023). We thank R. M. Ewers for contributing trait data. In Brazil, we also thank the Reserva Particular de Patrimonio Ùnio Natural de Serra das Almas, Área de Proteção Ambiental da Chapada do Araripe and Floresta Nacional do Araripe-Apodi for granting research access, and ONF Brazil, for providing data access from the Private Natural Reserve.

**Author contributions** J.A.-G. conceived the study, designed, performed and obtained funding for the analyses and wrote the first draft of the paper. S.W.R. and Y.M. contributed to the main ideas of the study. X.D. and H.Z.-Z. performed spatial analyses. Y.M. conceived and implemented the GEM network, obtained funding for most of the GEM trait field campaigns and commented on earlier versions of the manuscript. O.L.P. conceived and implemented ForestPlots.net together with T.R.B., obtained funding for most of its development, management, and support to tropical American partners, and commented on earlier versions of the manuscript. J.J.C.-R conceived and implemented the MONAFOR network, which is maintained and continues development also with support from M.G.N.-M and J.A.G. H.t.S., E.T., J.J.C.-R., A.F.G., S.M., J.K., S.F., A.F.R., G.W., C.A.J., K.A., S.A.-B., C.A.S., E.A.d.O., D.R. A.A., E.A.-D., G.P.A., T.R.B., M.B., L.P.B., E.B., L.B., D.B., K.B., R.B.d.L., S.B., J.C.D., D.C., H.C.d.L., L.C., L.A.C., N.C.C.d.S.P., A.C.d.S.Z., R.J.d.S., R.d.S.A.d.S., M.d.A.I., T.C.D.S.O., B.D., G.D., K.G.D., D.J. R., M.E.-S., L.F.S., T.F.D., J.F., M.F.S., C.A.J.G., B.H., K.J.J., S.K.A., A.K.S., B.K., W.F.L., M.L.D., W.E.M., E.M.C.-F., R.M.d.S., A.G.M., M.S., B.H.M.-J., R.E.M., D.L.M.V., T.M., W.M., P.M., M.M.M.d.S., P.S.M., R.M., M.G.N.-M., B.N., J.O.S., I.O.M., P.J.F.P.R., C.P.d.O., L.P.Z., C.A.P., V.P., C.A.Q., M.R.-M., T.R., G.R.-T., C.R., N.S., R.S.B., B.S.M., A.S., P.S.R., A.E.S.F., Q.S.G., T.S., D.S.-T., M.J.P.S., M.S., W.T.V.S., Y.A.T., P.T.S., M.T.N., E.V., I.C.Z.-B., M.R.D., S.T., M.P, V.B., S.d.P.C.C., L.J.T.W., E.Z., J.A.Z., D.F.R.P.B., M.S., J.C., B.J.E., J.B., O.L.P. and D.A.C. participated in or coordinated vegetation, trait data and/or soil data collection or processed field data, and commented on and approved the manuscript.

**Competing interests** The authors declare no competing interests.

**Additional information**
**Correspondence and requests for materials** should be addressed to Jesús Aguirre-Gutiérrez.

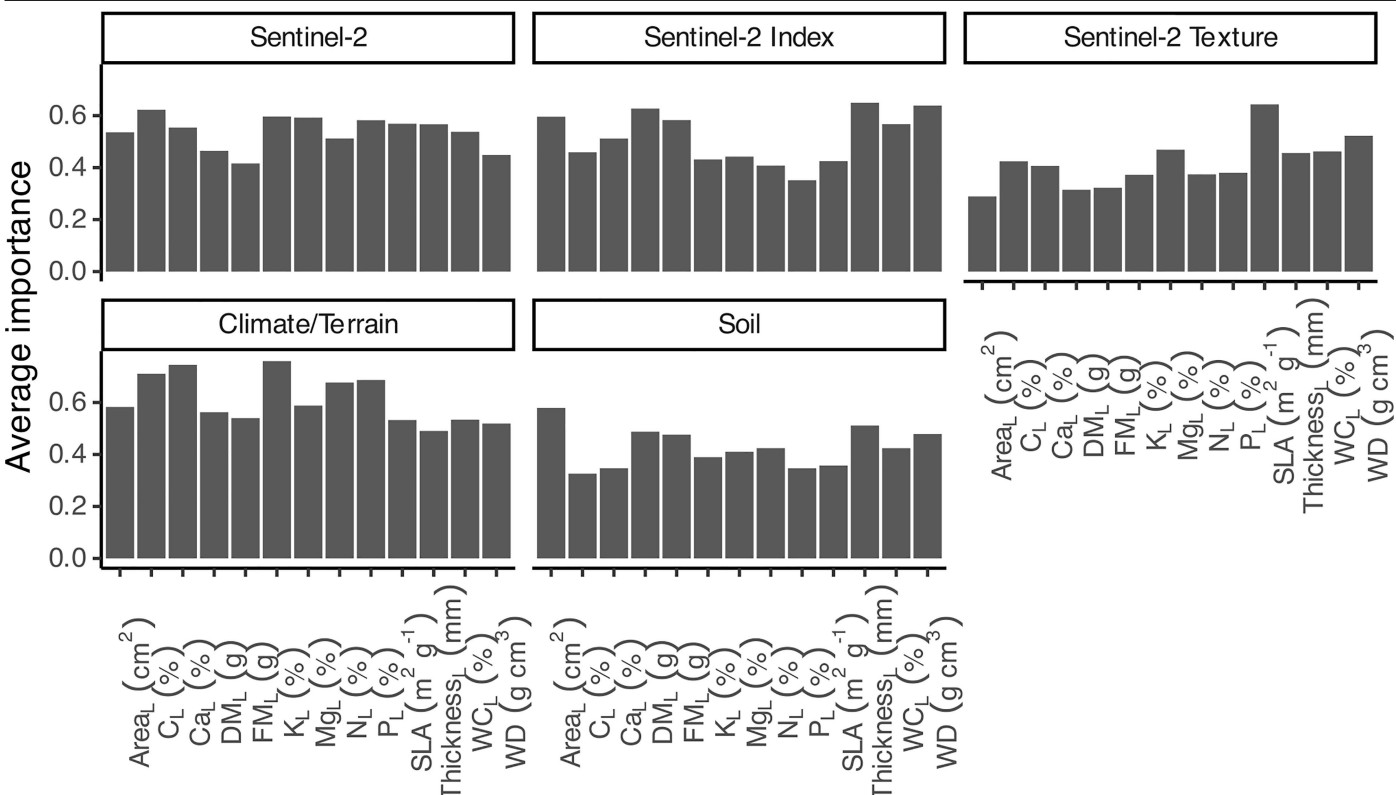

**Extended Data Fig. 1 | The importance of spectral data, vegetation indices, canopy texture parameters, climate, terrain and soil conditions for model prediction of each plant trait.** $Area_L$: leaf area, $C_L$: leaf carbon concentration, $Ca_L$: leaf calcium concentration, $DM_L$: leaf dry mass, $FM_L$: leaf fresh mass, $K_L$: leaf potassium concentration, $Mg_L$: leaf magnesium concentration, $N_L$: leaf nitrogen concentration, $P_L$: leaf phosphorus concentration, SLA: specific leaf area, $Thickness_L$: leaf thickness, $WC_L$: leaf water content, WD: wood density (see Extended Data Table 1 for a description of the trait used). The importance of each variable for each trait can be seen in Supplementary Figs. 1–13. The importance values were obtained from the RF models.

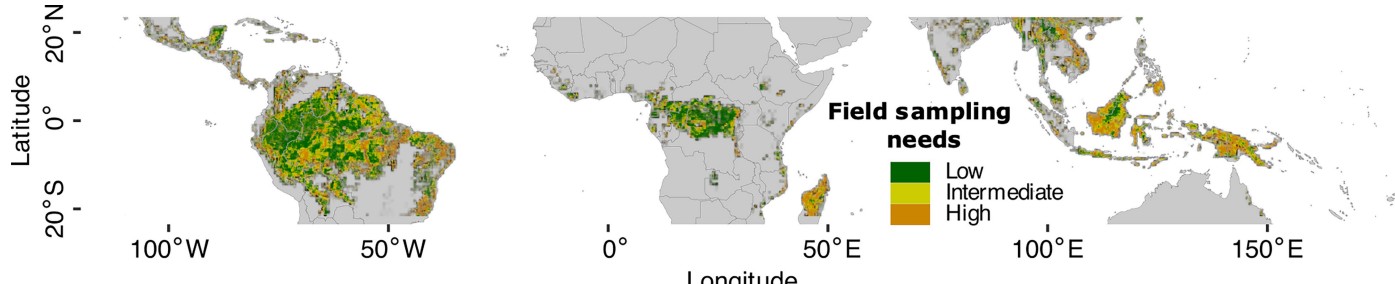

**Extended Data Fig. 2 | Predicted distribution of field sampling needs.** The map shows the locations where higher standard error of predictions of CWM trait values are found with orange showing high, yellow showing intermediate and green showing low sampling needs. The map was obtained by standardizing each CWM standard error (s.e.)-mapped prediction from 0 to 1 and obtaining an average value of the sum of those standardized SE maps. From this final field sampling needs map, we calculated the areas belonging to the lowest, middle and highest 33 percentiles and classified these as 'Low', 'Intermediate' and 'High' respectively. This final map could aid in generating field sampling priorities for the traits used in this study.

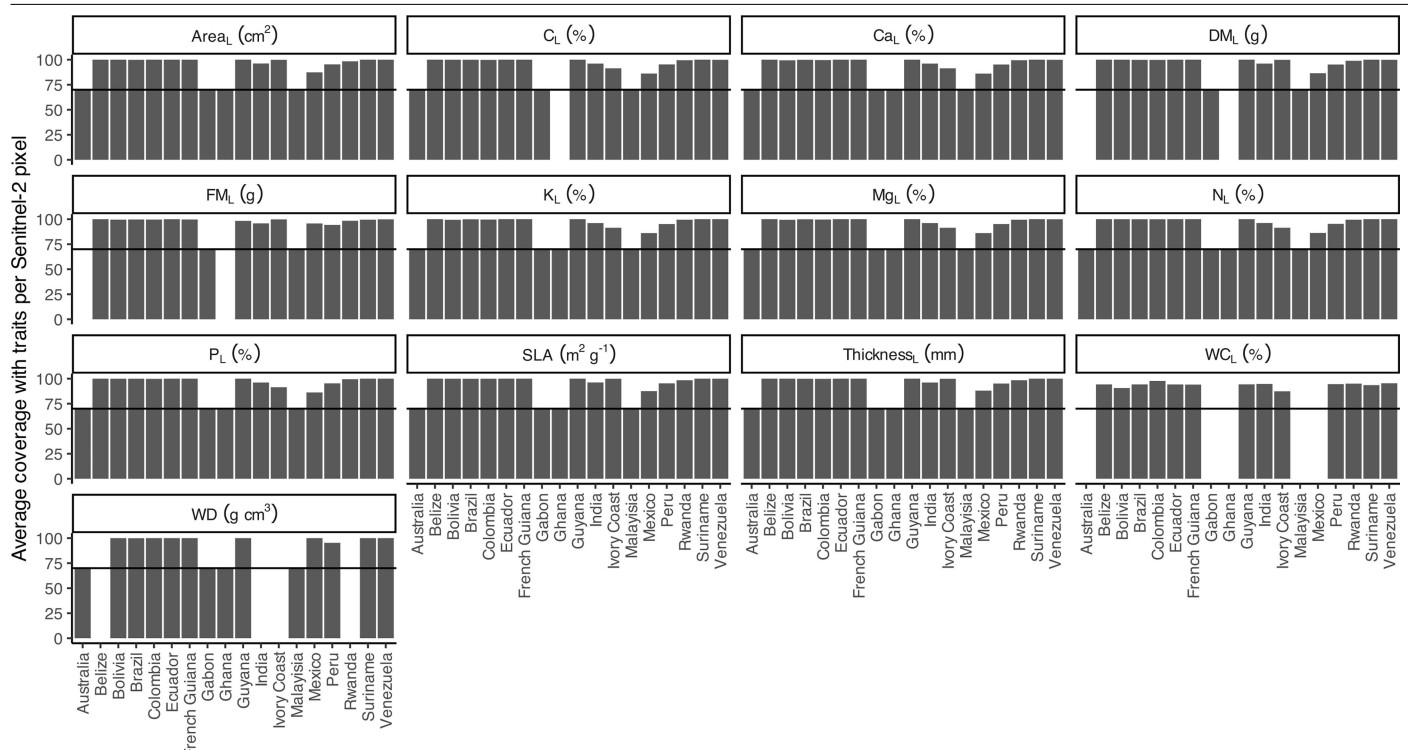

**Extended Data Fig. 3 | Percentage area covered by traits at the pixel level.**
Pixels had a minimum of 70% of the trees' basal area covered with trait data to enter the analysis. As shown, in several cases we reached higher than 70% basal area coverage at the pixel level. $Area_L$: leaf area, $C_L$: leaf carbon concentration,

$Ca_L$: leaf calcium concentration, $DM_L$: leaf dry mass, $FM_L$: leaf fresh mass, $K_L$: leaf potassium concentration, $Mg_L$: leaf magnesium concentration, $N_L$: leaf nitrogen concentration, $P_L$: leaf phosphorus concentration, SLA: specific leaf area, $Thickness_L$: leaf thickness, $WC_L$: leaf water content, WD: wood density.

**Extended Data Table 1 | Plant functional traits modelled and predicted across the tropics**

| Trait | Units | Description |
|---|---|---|
| Area (Area$_L$) | cm$^2$ | Area of the leaf as measured in one of its sides |
| Fresh mass (FM$_L$) | g | The fresh mass of a leaf |
| Dry mass (DM$_L$) | g | The dry mass of a leaf |
| Specific leaf area (SLA) | m$^2$ g$^{-1}$ | Area of the leaf as measured in one of its sides and divided by the dry mass |
| Thickness (Thickness$_L$) | mm | The thickness of a fresh leaf |
| Wood density | g cm$^3$ | Density of the wood of a tree |
| Water content (WC$_L$) | | Percentage water in the leaf relative to dry and fresh mass |
| Calcium content (Ca$_L$) | | |
| Carbon content (C$_L$) | | |
| Magnesium content (Mg$_L$) | % | |
| Nitrogen content (N$_L$) | | Leaf chemistry as concentration per unit dry leaf mass |
| Phosphorus content (P$_L$) | | |
| Potassium content (K$_L$) | | |

**Extended Data Table 2 | Description of the vegetation plots used across the tropical forests and their abiotic characteristics**

| Country | N | N Pixels | Area (ha) | MCWD (mm) | | Tmax (C) | | Slope (degrees) | |
|---|---|---|---|---|---|---|---|---|---|
| | | | | Average | CV | Average | CV | Average | CV |
| Australia | 4 | 403 | 4.03 | 62.95 | 12.55 | 26.26 | 3.23 | 16.15 | 54.97 |
| Belize | 10 | 1238 | 12.38 | 36.83 | 20.64 | 29.20 | 3.91 | 6.66 | 92.30 |
| Bolivia | 47 | 955 | 9.55 | 79.04 | 46.40 | 31.16 | 3.06 | 4.71 | 95.04 |
| Brazil | 324 | 20044 | 200.44 | 49.39 | 43.81 | 31.22 | 6.31 | 5.82 | 88.23 |
| Colombia | 39 | 3124 | 31.24 | 48.62 | 100.56 | 27.71 | 18.21 | 12.33 | 83.44 |
| Ecuador | 10 | 500 | 5 | 1.94 | 200.85 | 30.07 | 2.65 | 6.02 | 56.48 |
| French Guiana | 34 | 4230 | 42.3 | 48.93 | 1.15 | 29.98 | 0.03 | 8.03 | 72.93 |
| Gabon | 3 | 464 | 4.64 | 33.96 | 12.22 | 29.71 | 2.18 | 5.80 | 32.00 |
| Ghana | 4 | 620 | 6.2 | 61.80 | 41.50 | 30.34 | 0.68 | 5.20 | 28.77 |
| Guyana | 15 | 717 | 7.17 | 54.05 | 10.40 | 31.26 | 0.47 | 3.95 | 43.20 |
| India | 2 | 157 | 1.57 | 61.70 | 23.55 | 24.87 | 14.38 | 15.14 | 50.70 |
| Ivory Coast | 24 | 10493 | 104.93 | 101.55 | 0.18 | 31.41 | 0.03 | 4.79 | 56.70 |
| Malayisia | 8 | 976 | 9.76 | 17.05 | 6.34 | 29.44 | 1.74 | 14.40 | 63.42 |
| Mexico | 1140 | 28482 | 284.82 | 108.56 | 25.06 | 28.68 | 14.14 | 10.25 | 88.99 |
| Peru | 92 | 4912 | 49.12 | 31.08 | 103.06 | 27.94 | 15.62 | 11.57 | 88.21 |
| Rwanda | 13 | 881 | 8.81 | 53.74 | 25.63 | 21.31 | 8.00 | 17.05 | 43.18 |
| Suriname | 4 | 154 | 1.54 | 57.11 | 2.88 | 31.62 | 0.26 | 3.87 | 52.43 |
| Venezuela | 41 | 1605 | 16.05 | 69.59 | 60.60 | 31.14 | 11.23 | 6.08 | 93.13 |

N, number of vegetation plots; N Pixels, number of Sentinel-2 satellite pixels used; area (ha), planimetric pixel area used; MCWD, mean maximum climatic water deficit; Tmax, average maximum temperature; slope, average terrain slope. The average and coefficient of variation (CV as a percentage) are given for each climatic variable and were calculated using a climatology of the last 30 years (1988 and 2017). The climate data were extracted from the TerraClimate dataset[34] and the slope was derived from the Shuttle Radar Topography Mission (SRTM; www.earthdata.nasa.gov/sensors/srtm).

# Reporting Summary

## Statistics

For all statistical analyses, confirm that the following items are present in the figure legend, table legend, main text, or Methods section.

| n/a | Confirmed | |
|---|---|---|
| ☐ | ☒ | The exact sample size (*n*) for each experimental group/condition, given as a discrete number and unit of measurement |
| ☐ | ☒ | A statement on whether measurements were taken from distinct samples or whether the same sample was measured repeatedly |
| ☐ | ☒ | The statistical test(s) used AND whether they are one- or two-sided *Only common tests should be described solely by name; describe more complex techniques in the Methods section.* |
| ☐ | ☒ | A description of all covariates tested |
| ☐ | ☒ | A description of any assumptions or corrections, such as tests of normality and adjustment for multiple comparisons |
| ☐ | ☒ | A full description of the statistical parameters including central tendency (e.g. means) or other basic estimates (e.g. regression coefficient) AND variation (e.g. standard deviation) or associated estimates of uncertainty (e.g. confidence intervals) |
| ☐ | ☒ | For null hypothesis testing, the test statistic (e.g. *F*, *t*, *r*) with confidence intervals, effect sizes, degrees of freedom and *P* value noted *Give P values as exact values whenever suitable.* |
| ☐ | ☒ | For Bayesian analysis, information on the choice of priors and Markov chain Monte Carlo settings |
| ☒ | ☐ | For hierarchical and complex designs, identification of the appropriate level for tests and full reporting of outcomes |
| ☐ | ☒ | Estimates of effect sizes (e.g. Cohen's *d*, Pearson's *r*), indicating how they were calculated |

*Our web collection on statistics for biologists contains articles on many of the points above.*

## Software and code

Policy information about availability of computer code

| Data collection | No software was used to collect data |
|---|---|
| Data analysis | All analyses were carried in Google EarthEngine v. 2024 and the R statistical environment R version 4.0.5 (2021-03-31). RandomForest v. 4.7-1.1, Ranger V. 0.15.1, Princomp v. 4.3.1 , Funspace v.0.2.2 |

For manuscripts utilizing custom algorithms or software that are central to the research but not yet described in published literature, software must be made available to editors and reviewers. We strongly encourage code deposition in a community repository (e.g. GitHub). See the Nature Portfolio guidelines for submitting code & software for further information.

## Data

Policy information about availability of data

All manuscripts must include a data availability statement. This statement should provide the following information, where applicable:
- Accession codes, unique identifiers, or web links for publicly available datasets
- A description of any restrictions on data availability
- For clinical datasets or third party data, please ensure that the statement adheres to our policy

To comply with the original data owners' requirements, the plant functional traits and vegetation census data that support the findings of this study are available from their sources, GEM 31 at gem.tropicalforests.ox.ac.uk, and ForestPlots 32, 61, 62, www.ForestPlots.net and Diaz et al. 33. Given data sovereignty from the original data owners raw data on vegetation censuses and trait data are not publicly available but can be requested by contacting all researchers through the

## Human research participants

Policy information about studies involving human research participants and Sex and Gender in Research.

| | |
|---|---|
| Reporting on sex and gender | N/A |
| Population characteristics | N/A |
| Recruitment | N/A |
| Ethics oversight | N/A |

Note that full information on the approval of the study protocol must also be provided in the manuscript.

# Field-specific reporting

Please select the one below that is the best fit for your research. If you are not sure, read the appropriate sections before making your selection.

☐ Life sciences　　　☐ Behavioural & social sciences　　　☒ Ecological, evolutionary & environmental sciences

For a reference copy of the document with all sections, see nature.com/documents/nr-reporting-summary-flat.pdf

# Ecological, evolutionary & environmental sciences study design

All studies must disclose on these points even when the disclosure is negative.

| | |
|---|---|
| Study description | In this manuscript, we apply novel approaches encompassing field ecology and satellite remote sensing to produce and interpret for the first time spatially explicit predictions of 13 plant canopy functional traits covering the tropical forests distributed across the Americas, Africa, South East Asia and Australia. To tackle our questions, we field collected a comprehensive and unique pantropical dataset of highly relevant morphological, structural and leaf chemistry plant functional traits and linked them to >1800 unique plots with vegetation census which are recorded in across tropical forests. |
| Research sample | >1800 permanent plots |
| Sampling strategy | The sampling strategy was to collect plant functional traits for species that occupied at least 70% of the basal area of the vegetation plot and gather census data for as many undisturbed vegetation plots as possible. |
| Data collection | Local collaborators in each country and other co-authors from this manuscript collected the vegetation and census data. All vegetation census data is saved in the ForestPlots.net database (Leeds University) and the functional traits in the GEM traits database from the University of Oxford. |
| Timing and spatial scale | Most of the vegetation plots are of 1ha and their census time varies per region as these have been collected at different points in time but all censused in the last 5-10 years. The plant functional traits have been collected in most of the censused vegetation plots and covering at least 70% of the species basal area. The functional traits have been collected in tropical forests across the the four tropical continents. |
| Data exclusions | No data were excluded. |
| Reproducibility | Attempts to reproduce the analysis were successful. |
| Randomization | This is not relevant as we collected as much information as possible to make our analysis robust and did not need to randomize our samples for our analysis. |
| Blinding | Blinding was not relevant for our analysis. |

Did the study involve field work?　　☐ Yes　　☒ No

# Reporting for specific materials, systems and methods

We require information from authors about some types of materials, experimental systems and methods used in many studies. Here, indicate whether each material, system or method listed is relevant to your study. If you are not sure if a list item applies to your research, read the appropriate section before selecting a response.

## Materials & experimental systems

| n/a | Involved in the study |
|-----|------------------------|
| ☒ | Antibodies |
| ☒ | Eukaryotic cell lines |
| ☒ | Palaeontology and archaeology |
| ☒ | Animals and other organisms |
| ☒ | Clinical data |
| ☒ | Dual use research of concern |

## Methods

| n/a | Involved in the study |
|-----|------------------------|
| ☒ | ChIP-seq |
| ☒ | Flow cytometry |
| ☒ | MRI-based neuroimaging |

