## [Peer Review File · Nature]

Canopy functional trait variation across Earth's tropical forests

Corresponding Author: Dr Jesús Aguirre Gutiérrez

Version 0:

Reviewer comments:

Referee #1

(Remarks to the Author)

This manuscript presents an original and unique dataset from the GEM project investigating functional traits of tropical forests around the globe, coupled with fine scale (100m² pixel) spectral remote sensing data across tropical forests. The objective as established in the introduction is to ameliorate global biosphere models by providing global predictions for spatial distribution of tropical forest canopy tree traits related to key ecosystem services. In so doing, the manuscript also treats classic questions regarding how environmental factors shape tree trait distributions across tropical forests - although I believe the framing of these objectives and related predictions could be set up more comprehensively in the introduction.

The most compelling piece of the dataset involves relatively consistent coordinated sampling across three continents, providing the basis for standardized comparisons for the traits measured. Indeed, the storyboard focuses much attention on differences among the continents as a major first result of this work. I agree this comparison is original and provides some interesting results.

A second objective integrates data on climate and soil to investigate factors controlling the spatial distribution of traits within and among continents. The authors set this up as a major original contribution, but in so doing they gloss over some of the important work (which they cite but without complete treatment) that has already provided a foundation for trait-environment relationships, particularly within Neotropical forests. I believe the work here across continents represents an original contribution but that it can be compared more comprehensively to existing studies presenting complementary datasets. Here, a limited amount of raw data were translated to pixel level CWM which were then used to assemble global maps. Therefore, the value of these maps and the interpretations of these data hinge completely on the validity of these predictions.

The third embedded objective involves the link between field measured traits and spectral data to scale up to pixel-level predictions of functional trait values across global tropical forests. This objective is ambitious, and as argued would help to improve global biosphere models. However, I am not convinced that this current analysis represents the best means by which we can address this objective at this time.

A major reservation involves the rather limited datasets presented here. Given the ambitions, I wonder why complementary and in most cases published datasets were not integrated to expand the analyses, at least as part of the validation procedures. Yes, more than 2000 individuals were measured here, but this is really not a large sample size relative to the global scale addressed. And many of the measured plots cover a very limited surface area (0.1ha has very few canopy trees in most of these forests). Given the traits measured exist in publicly available datasets, many of which are linked to plot level data at similar scales, I am curious as to why more data were not included given the study's ambition? I strongly suggest that additional data be considered to strengthen the analyses and interpretations here.

A second reservation I have involves the validation and uncertainty approach, especially given the limited data coverage. I appreciate that four distinct methods were used in validation procedures. Nevertheless, I found the presentation of these procedures rather limited. The authors cite code for a supplementary table 5, but this was not available in the supplementary documents I had for review. I believe a more complete description of the procedures is warranted. I also again strongly suggest that additional data might be used in validation procedures.

A minor sidebar discussion involves the choice and presentation of functional traits. Ideally, everyone in this field would be measuring more traits that have more direct links to processes of interest - here especially response to soil fertility and a changing climate, eg heat and drought tolerance. However, the tradeoff for time to measure such traits at this impressive scale poses a considerable challenge. I believe the justification for the presented traits relative to objectives of global biosphere models merits further discussion. Here, rather than insisting on 11 traits, I would suggest exploring the relationships among these traits and treating synthetically the major axes (eg morphology, chemistry), which appear to be responding differently to the environment. Presenting all eleven traits makes the reading somewhat onerous and redundant; instead, one could choose 1-2 traits that represent each major axis and placing others in appendices.

Referee #2

(Remarks to the Author)

Aguirre-Gutierrez et al. presented a set of unique and interesting maps of plant functional traits across the tropics. I am quite intrigued by this manuscript, as it offers many exciting hypotheses that are worth pursuing by future studies. Meanwhile, I am 100% not sure whether Nature is the best place to publish this manuscript.

Overall, I believe that this manuscript presents cutting-edge science in the fields of plant ecology and tropical ecology. The authors utilized a novel and peer-reviewed method (Aguirre-Gutierrez et al. 2021) to generate maps of plant traits from large-scale satellite, climate, topography, and soil datasets. Their ground data came from plants from four tropical continents. Some of their maps are reasonably accurate, especially those for foliar chemical traits. I also admire the authors for describing their maps as work in progress and committing to improve the maps in the future. Again, I have no doubt that this work is top-notch and would likely be well cited.

I have some hesitation for recommending this to be published in Nature. One reason is that this manuscript does not present a coherent story, in my opinion. So many regional and continental trends are presented for a large number of plant traits. Some trends are interesting and consistent with past literature (e.g., east-west gradient of foliar P across Amazonia), while some are head-scratching, including some differences between continents. I feel a bit overwhelmed and cannot see a coherent story emerging from this paper. In addition, this paper is relatively long with almost 350 lines in the main text. Personally, I think this work can benefit from a traditional, long format (e.g., Ecology Letters) that allows the authors to fully articulate their points.

I also have some questions about their methodology. I think there are some artifacts in their maps of plant traits, including leaf area, thickness, C%, and water content. Specifically, the map values show irregular discontinuities along several straight lines, which likely are between satellite swaths. These artifacts are easier to spot in supplementary figures in Amazonia. I wonder to what extent these artifacts contribute to the reported trends? It also puzzles me how this work is different from Aguirre-Gutierrez et al. 2021. It appears that both works shared the same methodology and ground testing data.

Specific comments:

Line 71, regarding the geographical nuance, with the current model accuracy, it is hard to tell whether some of the nuances are real or just artifacts.

L126, 2,430 individuals represent impressive work. But it is still a relatively small number given the authors' interest in building a global-scale map. As they discovered, their results are highly sensitive to leaving one full vegetation plot out (L271). Since there are 47 plots (L356), is it possible that the current maps are biased in representing the entire tropics? Comments?

L288, these P trends are interesting. If leaf P declines with increasing water deficit, then dry forests should have lower leaf P than wet forests. However, this would contradict the finding from L224. Comments?

L333, could the authors specify their most meaningful/interesting/significant trait-environment relationships? I see many relationships presented but cannot seem to grasp their importance.

Supplementary Figure 7, no predicted vs. observed plot.

Referee #3

(Remarks to the Author)

In their paper, the authors present a pan-tropic mapping of 12 canopy functional traits obtained by training a random forest model on variables issued from a number of geospatial data layers, mostly from the Sentinel 2 satellite and mid-scale climate variables. The method is taken straight from [12].

They train their models with pixels issued from 47 plots located in the different continents. Once the maps obtained, they analyse the visual patterns thanks to knowledge from previous literature, relate observations between the different patterns and continental differences.

What I found interesting in the paper is the actual creation of the maps. These maps do not exist yet and give a first view of functional traits at the global (= pantropical) scale. This extend current knowledge, which is localised (regional, at best country) maps of one trait at a time. So in this respect the maps are of value.

As a remote sensing and machine learning specialist, I found some flaws in the technology used. Using a random forest for mapping with remote sensing is a very classical way of doing such mapping and recent works in vegetation mapping showed the superiority of neural network based approaches. I understand this is a matter of personal taste and RF vs NN is not a healthy debate per se (it is perfectly ok not to use a neural network), but I am worried by the low performance of the

model in validation. R2 is on average 50%, which is reasonable, but with large variations among variables and with a major design flaw: the fact that the selection of test samples is done at random over all plots, which makes the random forest classifier overfit, because it uses contextual variables (this is evident from the drop in performance when using a separate plot as validation data). This is something that would happen with every ML model, but I think it is fairer to report the results with the spatially disjoint validation, because it is the one that approaches more the quality of the global maps reported (one can expect these maps to have a R2 similar to those in the separate validation experiments as soon as we are out of the 47 plots). With that validation, R2 are more around 20-30%, which is problematic to then trust the patterns observed.

The scale of predictors also raises a question mark: authors are using predictors at completely different scales (10x10m for the satellite images, 30x30 for elevation, 4.6km for MCWD, etc), and then predict at 10m resolution. How would this affect predictions? And wouldn't this even more accentuate the data leaking between train and test? The texture data, computed on spatial neighbourhoods, and the 4km resolution data will forcedly introduce information from the test data into the training set. But despite the question of train /test split, I was surprised to see such a preponderance of the textural variables, since even if they are well documented and make sense in very high resolution imagery (e.g. to remove artefacts or highlight edges), at the resolution used here they would imply some kind of spatial autocorrelation in traits due to trees located tens (to hundred) of meters away? By the way, I could n't find the spatial window search of the GLCM computation, which would define such a spatial correlation range. As a final remark, GCLM features are somehow outdated, there is a vast literature of spatial filters that could be considered (BoVW, mathematical morphology, Fisher vectors, etc) that maybe could help improving performance.

For the standard deviation analysis, it is a good practice to run several instances of a model and record variation (btw the random forest does it by design by training the different (model) trees). I liked the claim that such uncertainty could lead future data acquisition campaigns, but I was wondering why authors did not push the exercise to the end and provide the map? It could be a simple (normalised) fusion of the uncertainty maps, followed by a discussion of "why" (see below) some areas are less accurately described than others in terms of predictors. Also, using the models trained with one plot out (rather than the random train-test split) would lead to more diverse models, which would in my opinion help in this search for new locations, even if it will also lead to larger uncertainties overall (because the models will disagree more).

Finally, I think the paper for the moment falls short of its own research questions. In the abstract authors claim as contributions:

1- present an analysis of how and why tropical forest vary in space. The "how" question is well addressed, in the sense that the authors discuss the 12 traits (one by one) in a dedicated paragraph and discuss what they observe in the maps. I am unfortunately not a forest specialist, so I cannot comment on the bio-geographic considerations made, but as an outsider I found the explanations of the patterns observed sometimes a little speculative. I would be a little careful in overanalysing them, since the accuracies are very low, so the patterns might simply be wrong. The "why" question is based on existing literature. This is fine, but I would argue that the "why" explanations are often given at the continental scale (figure 3) and therefore rely little on the actual maps, but more on the results on the existing plots. I would have loved to see some analysis of the patterns (correct and erroneous) on the plots left out for validation, which would have allowed to see in which way these 10m maps can advance the understanding the modeling of the traits locally (while being global). For the moment I see mostly observing some very large patterns and saying "yes, we expect Y because of reference X" and some hypothesis when the pattern are not what expected from literature. But again, what if those patterns are wrong?

2- provide insights into urgent questions about the effects of climate change. Without a temporal analysis I don't think this can be really tackled, and I cannot find this analysis in the paper. This would be extremely interesting (to run the model multiple years and study the variation of the traits), but it is not part of this work.

Minor questions:

In table 4 in the supplementary, from the filenames, it seems that the images are from 2015-2018, while in the text it is stated 2018-2021. Can you check?

I dont have access to Table 5 of the supplementary.

Page 13 describes results for the traits, which I would prefer in the main body of the paper.

Acknowledgments: Why giving access to the data upon request to the corresponding author only? This seems contradictory wrt to open science and makes all the access to data dependent on a single person.

I have never seen the formulation "(but see [ref])". Is it customary?

Is it normal that the main body of the text is a single section named "introduction"?

Referee #4

(Remarks to the Author)

Review Gutierrez et al., 2022 Nature

Dear Authors,

The manuscript presents a map of tropical forest plant functional traits using an extensive collection of field data and the new generation of optical satellite data. The authors map key functional traits and show that the distribution of the plant traits is mostly shaped by water availability, elevation, and local edaphic factors.

The authors also identify priority regions where further data collection or assimilation is needed to improve the knowledge of the spatial distribution of the selected functional traits in the tropics. The question is of great relevance, and the dataset is impressive. However, at this stage, I think the authors should provide more information and analysis about the robustness of the methodology to ensure that the results are based on a model suited for mapping the spatial patterns of the traits.

The use of Sentinel 2 is suited for these kinds of studies; however, I have a few comments. First, I had to go back to the article in RSE to understand the methodology better. Second, I suggest providing more details in the paper to understand the workflow better.

At line 388 the authors mention that the mean of the reflectance was used. The authors removed images with clouds, but it is known that Sentinel 2 cloud detection algorithm might fail for both small clouds and shades on the plots, and images affected by these issues might pass the quality control. Therefore, I suggest testing a more robust estimate of the reflectance, such as the median.

From line 427 onward: The authors aim at extrapolating their model beyond the training plots. The authors used good methods, and they have an excellent dataset. The authors evaluate the random forest model using different cross-validation approaches. The authors used both classical k-fold and spatial cross-validation, the latter very valuable for the assessment of the extrapolation capacity of the model. However, looking at the performance of the spatial cross-validation in table S3 I think the model's performance is not robust enough to be used for robust extrapolation. For some traits like N, the statistics are OK, not the best but at least they can be used for extrapolation purposes. However, for other traits (like SLA, area, P etc) cannot be fully considered robust. I invite the authors to provide more analysis on this point. For example, did the author try to benchmark the model against simple linear models or different machine learning methods? Would it be possible to apply a dimensionality reduction to the input parameters and improve the performance?

Line 430-431 – In my opinion, to provide this kind of uncertainty based on resampling would be better to use bootstrap (ideally with 500 runs but if the algorithm is too slow with at least 100 resample).

Figure 3 – at this stage is not possible to understand how solid those relationships are. The confidence intervals are missing. The underlying points are missing. This is important to evaluate the robustness of the regression, particularly at the lower and upper end of the distribution. The statistics of the different regression lines are missing and are needed to assess the robustness of the results discussed. Also, the authors analyzed the relationship between the mapped traits and other predictors. However, some of these predictors were used in the random forest model. Please clarify this aspect, and if this is the case, I think the results can be prone to spurious correlation. The authors might use different approaches, such as interpretable machine learning (partial dependence, SHAP values) to study the dependence of the traits to the underlying predictors. In general, I also feel that more details about the statistical analysis in the methods section are needed.

Minor points

-Line 58 I would not consider all these points as geographical factors

-Line 65-68 We highlight contrasts (unclear?). I would suggest the authors to develop this concept in the abstract better

-Global Biosphere Model and Earth system model are often used as synonyms, I suggest using only one of the

Lines 366-368: I suggest giving more details on which traits were calculated precisely. For example for the Nitrogen (N), do the author use eventually concentration, or N per area? Table 1 reports all this information. I suggest adding (see Table 1 for units and definitions) or providing the units and detailed descriptions in this section.

If I am not wrong, sentinel 2 data from GEE are not atmospherically corrected. While for the calculation of the vegetation indices, this might not be a problem, if the authors use the reflectance directly, it might be an issue. I suggest the authors verify the uncertainty related to using the data from GEE instead of using the ESA distribution workflow and related tools for correction. However, I am not a user of GEE, and I might be wrong, but I could not find more precise information online.

Version 1:

Reviewer comments:

Referee #2

(Remarks to the Author)

In the revised manuscript, the authors have significantly enhanced their work by incorporating an extensive dataset with more than 1800 vegetation plots. They have mapped the functional traits of the Earth's tropical forests using a novel method, presenting a coherent and convincing story that compares the functional syndromes of tropical forests across three continents. The addition of uncertainty analysis also brings new insights, particularly regarding future sampling efforts. The

authors have addressed my main concerns from the previous review regarding 1) the lack of a cohesive story and 2) a disconnect between observed patterns and their underlying mechanisms. The revised manuscript now offers an exciting map product that is likely to engage Nature's board readership. I have no major concerns. Below are a couple of minor questions.

In the functional richness analysis, to what extent are the observed differences across continents influenced by the differences in sampling efforts? I don't disagree with the authors' discussion of the differences in functional syndromes across continents, e.g., leaf phosphorus and other nutrients (lines 305-344). But the differences in sampling efforts are quite apparent (Figure 1). It would be beneficial if the authors could conduct a similar analysis using only the field data.

Figure 1C, many field plots are characterized by positive scores along PC3. They appear to be outliers relative to the environmental space. Comments?

Referee #3

(Remarks to the Author)

First I would like to thank the authors for their large efforts in revising the manuscript. It is particularly appreciated how the dataset was extended to 1800 plots, which makes the conclusions much stronger (but their geographical distribution also increases the risk of biases, I'll come back to that later).

As for the first round (I was reviewer 3), first a disclaimer: being a machine learning and remote sensing person, I cannot really comment on the biological significance of the results, and I leave my fellow reviewers comment on it. I will again focus on the technical aspects.

1. I really appreciated the increase in dataset size. I can see the results are more solid and also the variable selection make more sense and is less biased toward coarse variable (soil, texture and climate).

2. regarding different approaches to vegetation mapping. I see that my comments on neural network were not considered in the review. I think that given the prominence of AI in current research (and in particular deep learning) they deserve a tiny spot in your related works. There are now major works that are based on such technology (see eg. <https://www.nature.com/articles/s41559-023-02206-6.pdf> , <https://www.nature.com/articles/s41598-021-95616-0> , <https://www.nature.com/articles/s41477-021-01001-0>). It is ok that the authors don't want to compare (I would love to give it a try myself :)), but recent literature should still be acknowledged.

3. regarding spatial testing. Same for the spatial testing, there was no real response to my comments in the updated manuscript. I understand that random sampling gives better numbers, but for a work centered on large scale mapping, hence providing a map OUTSIDE of the plots area, random selection of pixel is not acceptable. Reading Table 3 in the supplementary, I see that the performances drop of 0.02-0.2 in terms of R^2 , with an average of 0.1. I think the drops are acceptable in terms of performance, especially given the global scope of the paper, but I really believe these are the results that should be reported in the main paper, and also used for all the maps and analysis. I understand that in Aguirre-Gutierrez et al. (2021) all the study was based on plot data, but here the objective is to evaluate a pantropical map that span across the plot data itself. The spatial testing results are the performance to be expected in the areas not in the plots and in this paper that is the majority.

4. Going further down this reasoning, the plots geographical distribution is very skewed to the Americas, which the authors discussed in the paper at line 94. Looking at plots in Fig 1A and 1B, there are still quite some environmental conditions not covered by the dataset, so I am still wondering if one should not try a more thorough analysis of the results using spatial splitting. For example, I would suggest to run some experiments where plots from the Americas are used for training and those from Asia and Africa are used for testing. This way we will see if there is a systematic bias in the predictions (by comparing the performance of these models to the "globally trained" one).

5. regarding the maps. I really like the maps and I think they are the added value of the paper (as stated in my previous review). And this is why I think the authors should make that one the main points of their discussion! What I mean is that the authors create these very high resolved maps of traits (10m, globally, it's massive), but then discuss mostly at regional or national scale. In the paper, we only see pan-tropical maps (e.g. fig 2 or in the supplementary), where one cannot really appreciate any of the details of specific regions. I am wondering if the same conclusions couldn't be reached with the plot data alone in many cases. What I am trying to say is that I would like to see cases where the patterns predicted allow some conclusions that we couldn't do with the plots data alone. I understand that some of the conclusions in the paper couldn't be done without those maps simply because there is no plot data, but having the discussion without visual examples hinders the appreciation I can have (again, as a remote sensing person). Maybe a figure with some examples would be enough to stress what is the actual plusvalue of these maps and the authors could provide a link to a map on Google Earth Engine where readers can zoom in and see the specific patterns.

6. regarding the PCA. I don't understand why the black outlines in Figure 4A and 4B are different. I thought they were outlining the data cloud of the whole dataset (otherwise those in the three subplots of 4B should also be different from each other). Can you explain? Also, I think subplot C should report more than the 2 first PC only. There is still ~40% of information in the subsequent PCs and I am sure there are different and interesting patterns to be found there. I know it doesn't help for the 2D plots in panels 4A and 4B, but For subpanel C I would go a bit deeper.

7. Regarding the uncertainty maps. I see that they are reported in terms of standard errors, to maintain their original units. This is nice, I was wondering if it would be possible to add a relative version (e.g. reported in number of standard deviations for all maps). This way, one could average them and provide a "global" score of uncertainty that could be used to search for the most interesting places to search (as stated at line 206). There are works going in this direction (e.g. active learning: <https://ieeexplore.ieee.org/stamp/stamp.jsp?arnumber=6185660>) and such a map could be of great interest to drive future definition of measurement plots. Now one would have to use and weigh in... 12 maps. Alternatively, one could count on how many maps a certain uncertainty threshold is surpassed and use those as "interesting" areas.

Very minor things:

- sometimes the abbreviations are in square parenthesis, sometimes in round parenthesis, please normalise
- can the uncertainty maps made as big as the prediction maps? I really appreciated that the prediction maps were given space in the supplementary material and I don't understand why the uncertainty maps are relegated into minuscule plots.

Referee #4

(Remarks to the Author)

The manuscript presents a map of 13 tree morphological, structural and chemical functional traits in the tropical pan-belt. The maps are based on a large collection of ground data, satellite information, terrain, climate and soil data.

The authors also analyse the functional diversity of tropical forests in the Americas, Africa and Asia. The authors conclude that tropical American forests are more functionally diverse than tropical African and Asian forests. African forests are the most functionally

African forests are more functionally diverse than tropical American and Asian forests. The authors then use an uncertainty analysis to show which regions should be prioritised for further data collection and discuss how this diversity should be incorporated into Earth system models.

Narrative:

The authors put a lot of emphasis on the fact that the new feature maps can be useful for models. I do not argue against this as I fully agree. However, I feel that there is little discussion of this. I think there should be more emphasis on this aspect (and perhaps reference to papers showing that averaging parameters in large grid cells is not ideal, while having this description of functional diversity and trait variability is important). Also, and this is more of a suggestion on how I would approach the discussion, I would put more emphasis and discussion on where diversity hotspots are located, if there are differences between protected and unprotected areas for example, and the conservation needs of these areas, rather than on the modelling part.

Line 92-96. I do not fully agree with the statement "While our vegetation plots are more abundant in the tropical forests of the Americas and it could be thought they mostly represent the environmental conditions in this region than in Africa and Asia, our principal component analysis (Fig. 1B and 1C) shows that with our pantropical sampling sites we cover the most prominent environmental conditions found across tropical forests in the world."

The PCA analysis in Figure 1 B and C shows that there are large areas of the data space that are not covered by the sampling sites. It is true that the areas of higher density are covered, but these are likely to be the Amazon. In order to prove what the authors claim, I propose to show on the biplots the points that are located in Africa and Southeast Asia.

The geolocation error of Sentinel 2 is about 10 m according to Gascon et al. (2017). Therefore, the selection of only 1 pixel is not ideal. The authors should acknowledge this aspect and they can test the use of a 3x3 window for the training, or include multiple traits and average within a larger windows. Here the reference.

Gascon, F.; Bouzinac, C.; Thépaut, O.; Jung, M.; Francesconi, B.; Louis, J.; Lonjou, V.; Lafrance, B.; Massera, S.; Gaudel-Vacaresse, A.; et al. Copernicus Sentinel-2A Calibration and Products Validation Status. *Remote Sens.* 2017, 9, 584. <https://doi.org/10.3390/rs9060584>

Line 128-136 - These statistics are fundamental to understanding how reliable the model is. It is unclear what exactly these statistics are (mean R2 across different cross-validation strategies?). In the methods the authors state: "The accuracy of the predictions was quantified by the explained variance using the R2". Is the R2 in cross-validation at all? This should be clarified, and I suggest using the cross-validation statistic to show the predictive accuracy of the model (see also comments below on the unbalanced dataset).

Also, at least a mean error (MAE) and a relative error (rMAE) should be included. The R2 does not provide any insight into the error and does not help to understand how statistically sound the result is. The authors calculate the OOB RMSE, but this is never reported. Nevertheless, the error statistics should be included in the cross-validation.

Moreover, given the large difference between the sample size in the Amazon and the sample size in the African and Asian continents, it would be important to design a cross-validation that allows understanding the confidence of the model in extrapolation in Africa. For example, the authors should test a training of the model in the Amazon and in Asia and predict the points in Africa and the same for Asia. Otherwise, the cross-validation, as currently designed, may be overoptimistic and suffer from the fact that the samples are so spatially unbalanced. Another option is to perform a balanced cross-validation. This can be done by maintaining the share of data from Amazon, Africa and Asia, in the training dataset, train the model and

predict on the unseen data points, but then test the performance and present them by continent.

The maps are made independently. For example, maps of N and P are produced without any constraint on the stoichiometric ratios that we know exist in the leaf sample. In my opinion, another aspect that should be evaluated to prove the robustness of the approach is to calculate maps of N/P and C/N ratios and evaluate if these ratios are in the ballpark of what is measured in the field.

The authors have analysed the uncertainty of the prediction, which is a great addition. However, I would suggest that the authors plot the trait maps (left column) and the trait uncertainty maps (right column). For each trait, I would keep the same colour scale for the trait map and the uncertainty map. This way the reader can see how large the uncertainty is compared to the trait values.

Version 2:

Reviewer comments:

Referee #3

(Remarks to the Author)

Once again, I would like to thank the authors for their large efforts in revising the manuscript. They make major efforts to include our comments! Thank you for adding the zoomed maps and the discussion on the per-continent evaluation. I think this makes the conclusions more credible, despite the (expected) drop in performance. Also I appreciate the extra discussion on the potential of using those traits maps in SDMs, I would have loved to have those in a recent study of ours (van Tiel et al., in case you are interested)!

Overall, for me the paper is in really good state, I mostly provide some minor, mostly aesthetic comments. As for the first two rounds, I will again focus on the technical aspects.

1. Since it is mentioned in the intro that there is a need for tools with high temporal resolution, I would appreciate a short paragraph in the discussion mentioning what would be needed to achieve that. I understand this is beyond the scope of this paper, but it would be nice for people willing to expand on this paper to know what would be necessary to turn this into a monitoring tool. Also, an estimate of the data and compute needs would be great if possible.

2. AI / deep learning paragraph (lines 331-342): I appreciate the addition of the paragraph, thank you! There are still a couple of imprecisions, that I would like to point out (I understand the authors are not deep learning specialists, it is really to improve the text):

- "AI beyond machine learning approaches": for me, "AI" is more of a fashionable term, nowadays corresponding to the usage of ML models (and the most often involving deep learning). Maybe you want to rephrase in to something like "testing the capabilities of large machine learning models, possibly involving deep learning, and enabling the capabilities of using data from multiple sources. A reference for that could be the SATBird paper (<https://satbird.github.io>).

- "Almost in real time": well, I would be a bit careful in overpromising that. Large scale DL model come with strong computational needs and also will require a lot of data downloads and preprocessing (see also my comment in 1. above). I would probably not mention this and simply state that we can have a temporally-aware, multi-sensor series of interesting models based on recent powerful advances in AI.

- I love your last point about not forgetting ecological models! I think this is where the field of hybrid DL for ecology should go!

There are a couple of examples around (for example taxonomy-aware models

<https://www.sciencedirect.com/science/article/pii/S0924271621002641> or models for structured biological knowledge

<https://imageomics.github.io/bioclip/>). Maybe worth adding those to the discussion?

As a personal add-on, I am looking forward to read about your new results with deep nets.

3. line 370: please add a comma after "traits", I think it would ease reading

4. line 454: "capabilities against overfitting and variable inputs"  "capabilities against overfitting and for its flexibility with respect to variations in the type and number of variable inputs"

5. line 462: "model"  "models" (or "model for each trait")

6. line 474: remove "at"

7. lines 480-484: I would put those after the paragraph ending at line 471, since they pertain to the traits prediction, not the uncertainty.

8. Supplementary table 7. I appreciate the authors making snippets of code available, but why not giving a link to a github with some sample code? This would have the advantage of potential updates and avoid obsolescence of your code.

9. Line 148: why leaving the "Link here"? I would have loved to check the online resource.

Thank you for all your efforts,
D. Tuia

‘Canopy functional trait variation across Earth’s tropical forests’

GENERAL COMMENTS FROM ALL REVIEWERS

R1-4 This ‘manuscript presents cutting-edge science in the fields of plant ecology and tropical ecology’... ‘I have no doubt that this work is top-notch and would likely be well cited’ (R2). The question is of great relevance, and the dataset is impressive (R1, R4) with ‘consistent coordinated sampling across three continents, providing the basis for standardized comparisons for the traits measured’, and with the cross-continental comparison being ‘original and provides interesting results’ (R1). The authors utilize ‘a novel and peer-reviewed method’ (R2) and create maps which ‘do not exist yet and give a first view of functional traits at the global (= pantropical) scale’ (R3).

Response

We thank the four reviewers for their supportive and positive feedback. It is truly encouraging to learn that you not only enjoyed reading the manuscript but also recognized the already high quality of our dataset. We appreciate your acknowledgment of our objectives, the impact of our maps, and the application of our previously published methodology. Your valuable feedback has been diligently incorporated into the updated version of the manuscript.

We acknowledge that, despite the general consensus on the high quality of our dataset, a common concern raised by most reviewers was the small sample size. Addressing this concern has been a focal point of our efforts, and we have taken substantial measures to enhance the dataset. We reached out to our local collaborators across tropical forests worldwide, resulting in a substantial increase in the number of vegetation plots. The revised manuscript now incorporates data from 1814 vegetation plots distributed across tropical forests on all four continents.

In response to your insightful comments, we have not only refined the presentation of the plant functional trait maps but also delved deeper into the investigation. Our analysis now extends beyond the mere exploration of plant trait distributions to encompass the examination of functional strategies, functional richness, and divergence across these diverse tropical forests.

To facilitate your review process, we have categorized the reviewers' comments into specific sections, providing detailed point-by-point responses below each section.

DATA USED

R1.2 Here, a limited amount of raw data were translated to pixel level CWM which were then used to assemble global maps. Therefore, the value of these maps and the interpretations of these data hinge completely on the validity of these predictions.

AND R1.4 A major reservation involves the rather limited datasets presented here. Given the ambitions, I wonder why complementary and in most cases published datasets were not integrated to expand the analyses, at least as part of the validation procedures. Yes, more than 2000 individuals were measured here, but this is really not a large sample size relative to the global scale addressed. And many of the measured plots cover a very limited surface area (0.1ha has very few canopy trees in most of these forests). Given the traits measured exist in publicly available datasets, many of which are linked to plot level data at similar scales, I am curious as to why more data were not included given the study’s ambition? I strongly suggest that additional data be considered to strengthen the analyses and interpretations here.

Response

We acknowledge and appreciate your concerns regarding the quantity of data used in our study. In response, we have undertaken significant efforts to address this matter. We have successfully gathered a comprehensive dataset comprising a total of 1814 vegetation plots. Each plot includes detailed information on the species present, their respective locations within the plot, and available plant functional traits. Notably, these 1814 plots are strategically distributed across tropical forests spanning all four continents and collectively cover an extensive area of nearly 800 hectares.

We assert that this augmentation represents a substantial advancement in the field, constituting what we believe to be the most extensive endeavour to date in mapping plant functional traits across tropical forests. Importantly, this initiative is distinguished by the utilization of high-quality, field-collected data, further enhancing the robustness and reliability of our findings.

R1.3 The third embedded objective involves the link between field measured traits and spectral data to scale up to pixel-level predictions of functional trait values across global tropical forests. This objective is ambitious, and as argued would help to improve global biosphere models. However, I am not convinced that this current analysis represents the best means by which we can address this objective at this time.

Response

Certainly, we concur with your perspective, and we have revised our manuscript to propose that the enhanced predictions of plant traits serve as a valuable tool, providing more pertinent information about the functional composition of canopies. This, in turn, can be utilized as input data for global biosphere models. We are confident that the new dataset we have compiled facilitates the generation of high-quality plant trait predictions suitable for incorporation into such models.

However, it is essential to emphasize that the true added value of incorporating these traits into global vegetation models necessitates thorough testing, a step we have been unable to undertake as these models did not previously exist, as noted by other reviewer(s). Consequently, recognizing this limitation, we express our intention to make our trait maps accessible to the broader ecological modelling community. This will enable researchers to assess the added value of these traits within the context of their research frameworks, fostering collaborative efforts to validate and refine the contribution of our findings.

METHODOLOGY

R1.5 A second reservation I have involves the validation and uncertainty approach, especially given the limited data coverage. I appreciate that four distinct methods were used in validation procedures. Nevertheless, I found the presentation of these procedures rather limited. The authors cite code for a supplementary table 5, but this was not available in the supplementary documents I had for review. I believe a more complete description of the procedures is warranted. I also again strongly suggest that additional data might be used in validation procedures.

Response

We have incorporated additional information in our manuscript detailing the procedures for assessing the accuracy of our predictions, providing comprehensive insights into the test approaches employed. We regret the absence of one of the tables in our previous version and assure you that all relevant material is now included for your comprehensive review.

Notably, this updated version leverages a significantly more extensive dataset, surpassing the scale of our previous iteration by several orders of magnitude. The incorporation of three rigorous testing methods, namely 10-fold cross-validation, an 80% training and 20% testing split, and Spatial Leave-One-Out Cross-Validation, reinforces the robustness of our results. This expanded dataset, coupled with the diverse testing approaches, substantiates the assertion that our findings are highly robust and inherently data-driven. This approach stands in contrast to other products published to date, further emphasizing the reliability and credibility of our results.

R2.3 I also have some questions about their methodology. I think there are some artifacts in their maps of plant traits, including leaf area, thickness, C%, and water content. Specifically, the map values show irregular discontinuities along several straight lines, which likely are between satellite swaths. These artifacts are easier to spot in supplementary figures in Amazonia. I wonder to what extent these artifacts contribute to the reported trends? It also puzzles me how this work is different from Aguirre-Gutierrez et al. 2021. It appears that both works shared the same methodology and ground testing data.

Response

The Bidirectional Reflectance Distribution Functions (BRDF) is a well-known factor affecting satellite imagery. Our efforts to enhance imagery using recently developed workflows, such as the one outlined in the provided link (<https://custom-scripts.sentinel-hub.com/sentinel-2/brdf/#>), have been explored. However, as acknowledged by the authors, these improvements do not consistently result in enhanced images, and ongoing research is dedicated to understanding and mitigating BRDF effects. Notably, our updated products, as illustrated in the mentioned traits, reveal minimal BRDF effects, and in several cases, no discernible impact. Part of our ongoing research involves developing Earth Engine algorithms specifically designed to more effectively address BRDF effects, although these algorithms are still in the developmental phase.

Regarding the extent to which BRDF effects drive the observed patterns, our models are now primarily influenced by reflectance data obtained from thousands of pixels in Sentinel-2 imagery, alongside pixel-level Community-Weighted Mean (CWM) trait values collected in the field. This combination leads us to believe that the patterns observed in the maps are primarily shaped by the dominant signal present in the data rather than BRDF effects, which tend to manifest more prominently at the edges of satellite imagery. Additionally, our study demonstrates how the observed trait distribution patterns align with field-collected data, conform to expectations based on recent literature, and exhibit anticipated variations across the study area.

In distinguishing our work from Aguirre-Gutierrez et al. 2021, we emphasize that our 2021 manuscript introduces a novel methodology, acknowledged by publication in a prominent remote sensing journal. However, this earlier work did not specifically focus on generating maps at a pantropical extent, analysing trait distributions, or investigating functional differences across dry and wet tropical forests. These aspects are central to the current revision of our manuscript, representing a significant advancement in our ecological understanding of the functional diversity within tropical forests.

R3.2 Using Machine Learning for modelling and how the accuracy assessment has been done. &
R4.1 did the author try to benchmark the model against simple linear models or different machine learning methods? Would it be possible to apply a dimensionality reduction to the input parameters and improve the performance?

Response

As you mention and we agree, there are several relevant algorithms and approaches for mapping biodiversity and plant traits. RF has demonstrated its efficacy in our peer-reviewed manuscript, where the modelling and mapping methods were published, and its application was benchmarked against other algorithms (glm and PLSR). The decision to use RF is supported by its consistently equal or superior predictive performance when compared to alternative methods, as evidenced by various relevant studies listed below. The algorithm's capability to model complex nonlinear relationships, its nonparametric nature, and its resilience against overfitting make it particularly well-suited for our study. RF's robustness, attributed to its bagging process and the incorporation of random feature selection, aligns with the challenges posed by diverse sets of covariates in our study. The algorithm's capacity against overfitting is especially valuable given the inclusion of several covariates in our models and the bagging process effectively mitigates potential overfitting concerns. Additionally, the careful selection of variables, based on their demonstrated impact on the distribution of the modelled traits, contributes to the model's reliability. While acknowledging the dimensionality of our dataset, we have applied dimensionality reduction in the early analyses, eliminating non-essential covariates from the dataset. This strategic approach further enhances the model's efficiency and interpretability.

Here some relevant references describing the use Machine learning Random Forests and showing their capacity and accuracy for predicting plant traits and biodiversity metrics at scale. We have also included these references in the new version of our manuscript:

*Aguirre-Gutiérrez, Jesús, et al. Pantropical modelling of canopy functional traits using Sentinel-2 remote sensing data. *Remote Sensing of Environment* 252 (2021): 112122.

*Boonman, C. et al. Assessing the reliability of predicted plant trait distributions at the global scale. *Global Ecology and Biogeography* 29.6 (2020): 1034-1051.

*Ali, A. et al. Machine learning methods' performance in radiative transfer model inversion to retrieve plant traits from Sentinel-2 data of a mixed mountain forest. *International Journal of Digital Earth* 14.1 (2021): 106-120.

*Cai, Lirong, et al. Global models and predictions of plant diversity based on advanced machine learning techniques. *New Phytologist* 237.4 (2023): 1432-1445.

*Thomson, E. R., et al. Multiscale mapping of plant functional groups and plant traits in the High Arctic using field spectroscopy, UAV imagery and Sentinel-2A data. *Environmental Research Letters* 16.5 (2021): 055006.

R3.3 Explaining the scale of the predictors used and the use of GLCM and their search window

Response

Our focus scale is 10x10 m, corresponding to the pixel size of the Sentinel-2 satellites, from which we derive the spectral data, a primary input for our models. Soil, topography, and climate data are also incorporated, all of which are resampled to match the 10x10 m pixel size. We acknowledge the inherent differences in spatial scales among predictors, especially in analyses spanning a pantropical extent. In the tropics, where conditions are more homogeneous, we anticipate minimal variations in climate values at the landscape level. Despite the ideal preference for plot-level soil data, it was unavailable for most vegetation plots, necessitating reliance on global soil datasets. Topography covariates were initially extracted at a 30x30 m pixel size and subsequently resampled to the 10x10 m resolution. Responding to your request, we have expanded and clarified our methodology, providing detailed descriptions of the datasets used, including an elaboration on the grey level co-occurrence matrix and neighbour distance, represented by a 9x9 cell neighbourhood as in our previous methodology manuscript from Remote Sensing of the Environment (reference below). Here's an excerpt of the new text:

‘The GLCM metrics are computed from a matrix that is spatially dependent. The co-occurrence matrix relies on the angular orientation and distance between adjacent pixels, illustrating the frequency of associations between a pixel and its neighbouring pixels. We applied a 9×9 pixel kernel window as in our previous published methodology 4 where texture results obtained using this kernel window exhibited a strong correlation with those from the smaller kernel window ($Cor = 0.94$, $P \leq 0.0001$). This choice proved sufficient to capture ample canopy contrast information during the modelling stage without incurring substantial computation time.’

Aguirre-Gutiérrez, Jesús, et al. "Pantropical modelling of canopy functional traits using Sentinel-2 remote sensing data." *Remote Sensing of Environment* 252 (2021): 112122.

R3.3 Including the maps of the uncertainty in predictions. -I liked the claim that such uncertainty could lead future data acquisition campaigns, but I was wondering why authors did not push the exercise to the end and provide the map?

Response

We agree that presenting the mapped predictions of uncertainty would be valuable for understanding areas where additional field campaigns are highly needed, potentially leading to significant improvements in our current models. Maps of the uncertainty predictions for all traits are now included in Supplementary Figures 15 and 16.

R4.2 The question is of great relevance, and the dataset is impressive. However, at this stage, I think the authors should provide more information and analysis about the robustness of the methodology to ensure that the results are based on a model suited for mapping the spatial patterns of the traits. The use of Sentinel 2 is suited for these kinds of studies; however, I have a few comments. First, I had to go back to the article in RSE to understand the methodology better. Second, I suggest providing more details in the paper to understand the workflow better.

Response

We are pleased that you found our dataset impressive in the first version of the manuscript and appreciate your endorsement of our approach utilizing Sentinel-2 imagery. In response to your

valuable suggestions, we have made extensive modifications to the manuscript, incorporating additional data, conducting diverse analyses, and enhancing the methodology descriptions throughout the text.

In these new analyses and text revisions, we not only explore the prediction of traits across tropical forests but also extend our investigation to examine the distribution of possible trait syndromes, functional trait richness, and functional trait divergence across continents. We trust that you will find our revised manuscript more self-explanatory, and we believe that the new analyses will be of particular interest to you.

R4.3 At line 388 the authors mention that the mean of the reflectance was used. The authors removed images with clouds, but it is known that Sentinel 2 cloud detection algorithm might fail for both small clouds and shades on the plots, and images affected by these issues might pass the quality control. Therefore, I suggest testing a more robust estimate of the reflectance, such as the median.

Response

We agree with your suggestion and we have re-run all analyses with our improved dataset and have used the median reflectance value.

R4.4 From line 427 onward: The authors aim at extrapolating their model beyond the training plots. The authors used good methods, and they have an excellent dataset. The authors evaluate the random forest model using different cross-validation approaches. The authors used both classical k-fold and spatial cross-validation, the latter very valuable for the assessment of the extrapolation capacity of the model.

Response

Thank you for your encouraging remarks and positive feedback about our approach.

R4.5 Line 430-431 – In my opinion, to provide this kind of uncertainty based on resampling would be better to use bootstrap (ideally with 500 runs but if the algorithm is too slow with at least 100 resample).

Response

Following your suggestion, we have now done something alike what you propose using the infinitesimal jackknife (Wager et al. 2014) and present our results in Supplementary Figures 15 and 16. We employ the infinitesimal jackknife estimate of variance due to its enhanced efficiency in utilizing preexisting bootstrap replicates compared to the jackknife estimator. This efficiency is attributed to its lower Monte Carlo variance, necessitating 1.7 times fewer bootstrap replicates than the traditional jackknife method. Within the infinitesimal jackknife framework, we investigate the impact on the statistic when each observation is individually down-weighted by an infinitesimal amount. Notably, the infinitesimal jackknife often yields more stable predictions than the conventional jackknife. Therefore, we adhere to a bootstrap approach, specifically tailored for Random Forest models, as recommended for its appropriateness in our context.

Wager, Stefan, Trevor Hastie, and Bradley Efron. "Confidence intervals for random forests: The jackknife and the infinitesimal jackknife." *The Journal of Machine Learning Research* 15.1 (2014): 1625-1651.

R4.5 Figure 3 – at this stage is not possible to understand how solid those relationships are.

Response

In response to your guidance and that of other reviewers, we have chosen to concentrate our manuscript on functional trait predictions, ecological syndromes across continents, and the exploration of functional diversity (richness and divergence) across continents. Consequently, the

original analyses associated with Figure 3 in the previous manuscript are not included in this revised version.

WRITING OF THE MANUSCRIPT AND FOCUS

R2.2 ...this manuscript does not present a coherent story, in my opinion. So many regional and continental trends are presented for a large number of plant traits. Some trends are interesting and consistent with past literature (e.g., east-west gradient of foliar P across Amazonia), while some are head-scratching, including some differences between continents. I feel a bit overwhelmed and cannot see a coherent story emerging from this paper. In addition, this paper is relatively long with almost 350 lines in the main text. Personally, I think this work can benefit from a traditional, long format (e.g., Ecology Letters) that allows the authors to fully articulate their points.

Response

We appreciate your input, and we have incorporated your comments into the new version of the manuscript. Recognizing the potential for improvement, we have redesigned all our analyses. Instead of delving into the individual description of each trait, our focus is now directed towards exploring the distribution of plant syndromes—variations in trait combinations—across continents. Additionally, we examine the variation in functional richness and functional divergence in tropical forests across continents.

The new version of our manuscript, based on a much richer dataset (1814 permanent vegetation plots), we believe, presents a more coherent story and concentrates on the most interesting and striking ecological outcomes. We hope you will find it of interest.

MINOR SPECIFIC COMMENTS

R1.6 A minor sidebar discussion involves the choice and presentation of functional traits. Ideally, everyone in this field would be measuring more traits that have more direct links to processes of interest - here especially response to soil fertility and a changing climate, eg heat and drought tolerance. However, the tradeoff for time to measure such traits at this impressive scale poses a considerable challenge. I believe the justification for the presented traits relative to objectives of global biosphere models merits further discussion. Here, rather than insisting on 11 traits, I would suggest exploring the relationships among these traits and treating synthetically the major axes (eg morphology, chemistry), which appear to be responding differently to the environment. Presenting all eleven traits makes the reading somewhat onerous and redundant; instead, one could choose 1-2 traits that represent each major axis and placing others in appendices.

Response

We appreciate your guidance and have implemented your suggestions in the new version of the manuscript. In this revised edition, we highlight the most striking trait patterns while emphasizing the relationships between them, particularly the emerging 'syndromes' across tropical forests.

Additionally, we delve into the analysis of functional richness and divergence stemming from these traits across tropical forests.

Following your recommendation, we have included more detailed predictions of all traits in the Supplementary Information. We hope that these new analyses and results prove to be of interest, and that the changes made to our manuscript based on your comments are suitable.

R2.4 Line 71, regarding the geographical nuance, with the current model accuracy, it is hard to tell whether some of the nuances are real or just artifacts.

L126, 2,430 individuals represent impressive work. But it is still a relatively small number given the authors' interest in building a global-scale map. As they discovered, their results are highly sensitive to leaving one full vegetation plot out (L271). Since there are 47 plots (L356), is it possible that the current maps are biased in representing the entire tropics? Comments?

L288, these P trends are interesting. If leaf P declines with increasing water deficit, then dry forests should have lower leaf P than wet forests. However, this would contradict the finding from L224. Comments?

L333, could the authors specify their most meaningful/interesting/significant trait-environment

relationships? I see many relationships presented but cannot seem to grasp their importance. Supplementary Figure 7, no predicted vs. observed plot.

Response

Line 71, 126: We fully agree with your observations. The nuances in the results may have been more uncertain when we initially used only 47 vegetation plots in our analyses. Taking your advice and that of other reviewers into serious consideration regarding the representativeness of the environment across the tropics with that limited dataset, we embarked on an extensive effort to enhance our work. We conducted our own fieldwork and collaborated with over 100 researchers, resulting in a substantial increase in our dataset. The current version incorporates data from 1814 vegetation plots, making it, to the best of our knowledge, one of the most significant efforts to map plant traits at a pantropical scale. We appreciate your comment, as it prompted us to reframe our analyses and bring forth a much richer dataset for this new version of our manuscript.

R3.4 Addressing the questions Why and How in the paper: ‘present an analysis of how and why tropical forest vary in space. The "how" question is well addressed’.... The "why" question is based on existing literature. This is fine, but I would argue that the "why" explanations are often given at the continental scale (figure 3) and therefore rely little on the actual maps, but more on the results on the existing plots.

Response

We have redefined the focus of our manuscript, providing detailed answers to the "how" and "why" across the various objectives outlined in this new version. Our attention is now squarely on the distribution of plant traits and syndromes, as well as on the richness and divergence of trait composition. For each of these aspects, our discussion is grounded in the presented results, aligning them with past and recent literature. We hope that you find these modifications valuable and that they effectively address your previous comments and concerns.

R3.5 Addressing the statement: ‘provide insights into urgent questions about the effects of climate change’.... Without a temporal analysis I don't think this can be really tackled, and I cannot find this analysis in the paper. This would be extremely interesting (to run the model multiple years and study the variation of the traits), but it is not part of this work.

Response

This is indeed a fantastic venue of research in which we are working on and it is certainly not addressed in our current manuscript. Therefore, following your advice, we have refrained from writing that phrase in our new version. For a separate research manuscript, we plan to undertake new analyses focusing on understanding possible changes in functional traits across time. These analyses will be based on the models presented in this version of our manuscript and satellite data. We look forward to sharing the results with you and the wider research community in the near future.

R3.6 In table 4 in the supplementary, from the filenames, it seems that the images are from 2015-2018, while in the text it is stated 2018-2021. Can you check?

Response

We have checked and have re-run all models using the satellite data for the period 2019-2022 and have updated the manuscript with this information.

R3.7 I dont have access to Table 5 of the supplementary.

Response

We want to confirm that, this time, all tables listed will be uploaded as a separate file along with the main manuscript on the journal's website.

R3.8 Page 13 describes results for the traits, which I would prefer in the main body of the paper.

Response

In our revised version, we have reorganised the order of the presentation, aligning the analysis, results, and discussion to showcase the trait distributions more effectively. We hope you find that these changes have improved the readability of the manuscript, emphasizing the most important results in their respective sections.

R3.9 Acknowledgments: Why giving access to the data upon request to the corresponding author only? This seems contradictory wrt to open science and makes all the access to data dependent on a single person.

Response

We acknowledge your comment, and in the revised version, we have provided a clear listing of the two core sources of the data. Access to the data is now available through the GEM and ForestPlots websites, eliminating dependence on any single author and accommodating current restrictions from data owners.

R3.10 I have never seen the formulation "(but see [ref])". Is it customary?

Response

This formulation is not used in the new version of the manuscript anymore.

R3.11 Is it normal that the main body of the text is a single section named "introduction"?

Response

In the new version of the manuscript, we included a heading for each section following the standard format of the journal.

R4.6 -Line 58 I would not consider all these points as geographical factors
-Line 65-68 We highlight contrasts (unclear?). I would suggest the authors to develop this concept in the abstract better
-Global Biosphere Model and Earth system model are often used as synonyms, I suggest using only one of the
Lines 366-368: I suggest giving more details on which traits were calculated precisely. For example for the Nitrogen (N), do the author use eventually concentration, or N per area ? Table 1 reports all this information. I suggest adding (see Table 1 for units and definitions) or providing the units and detailed descriptions in this section.

Response

Line 58: We have changed the text accordingly and have eliminated the term 'geographic'.
Line 65-68: This is not part of our text anymore.
Global Biosphere Model and Earth system model: We only use Earth System Model now.
Lines 366-368: We have included the traits information as suggested with a direct reference to Supplementary Table 1.

R4.7 If I am not wrong, sentinel 2 data from GEE are not atmospherically corrected. While for the calculation of the vegetation indices, this might not be a problem, if the authors use the reflectance directly, it might be an issue. I suggest the authors verify the uncertainty related to using the data from GEE instead of using the ESA distribution workflow and related tools for correction. However, I am not a user of GEE, and I might be wrong, but I could not find more precise information online.

Response

It would indeed be an issue, as you mentioned, if we were using top-of-atmosphere reflectance data (i.e., not atmospherically corrected). However, we want to highlight that we use surface reflectance

data, which undergoes atmospheric correction. The Sentinel-2 data is available in Earth Engine, and you can find a direct description of the product here:

<https://developers.google.com/earth-engine/datasets/catalog/sentinel-2>

Canopy functional trait variation across Earth's tropical forests

Responses to comments from the Editor and Reviewers

Dear Editor and Reviewers,

We thank you for taking the time to revise our manuscript. We have certainly learned a lot from your comments and based on them, we have greatly improved our manuscript. Below, we give a point-by-point response to all of your comments. We hope that our responses and our new version of the manuscript adequately address all your comments and questions.

Referee #2 was fully satisfied with the changes,

Response

We are glad Reviewer 2 is happy with the current manuscript.

Referee #3 feels that some of their previous concerns have not been engaged with, especially regarding concerns over spatial testing and addressing geographic biases in the plots. They would also like some more clarity on the uncertainty maps.

Response

We perhaps mistakenly did not directly reply to some of the comments, and now we have a point-by-point response to each one, especially about the model testing and uncertainty maps, as suggested. We have done one more test by spatial cross-validation using all data, and also using the approach for Africa and Asia only. This spatial approach is the one we now use along the manuscript to avoid any further confusion. We also present the maps of Standard Error of predictions next to the community weighted mean prediction maps.

Similarly Referee #4 feels that gaps and biases in the spatial sampling need additional acknowledgement, and they suggest some additional cross-validation may be required.

Response

We have addressed this comment as this is similar to the one from reviewer 3 above and have updated our manuscript accordingly. In short, we now do a spatial plot-leave one out cross validation approach to test the model accuracy as discussed above.

In addition, after consultation, this referee let us know that they felt that most of the major concerns raised by Referee #1 had been adequately addressed, with the exception of those concerns which overlapped with their own over the coverage of climate and environmental space.

Response

We are glad all concerns from reviewer 1 have been also adequately addressed and we hope all of its concerns are also addressed in this new version of our manuscript.

General comments

Referee #2

In the revised manuscript, the authors have significantly enhanced their work by incorporating an extensive dataset with more than 1800 vegetation plots. They have mapped the functional traits of the Earth's tropical forests using a novel method, presenting a coherent and convincing story that compares the functional syndromes of tropical forests across three continents. The addition of uncertainty analysis also brings new insights, particularly regarding future sampling efforts. The authors have addressed my main concerns from the previous review regarding 1) the lack of a cohesive story and 2) a disconnect between observed patterns and their underlying mechanisms. The revised manuscript now offers an exciting map product that is likely to engage Nature's board readership. I have no major concerns.

Referee

#3

First I would like to thank the authors for their large efforts in revising the manuscript. It is particularly appreciated how the dataset was extended to 1800 plots, which makes the conclusions much stronger.

Response

We are glad the reviewers found we significantly enhanced our work by incorporating an extensive dataset and that this effort is highly appreciated, also stating this makes the conclusions much stronger. It is also great to see the reviewers find our story 'coherent' and 'convincing' and that reviewer 2 has no further major concerns.

Referee #3

As for the first round (I was reviewer 3), first a disclaimer: being a machine learning and remote sensing person, I cannot really comment on the biological significance of the results, and I leave my fellow reviewers comment on it. I will again focus on the technical aspects.

Response

We thank you for your great input that from all perspectives and especially from the Remote Sensing component has helped us to improve our manuscript greatly. We hope you find this new version of the manuscript and our responses below complete and satisfactory.

1. I really appreciated the increase in dataset size. I can see the results are more solid and also the variable selection make more sense and is less biased toward coarse variable (soil, texture and climate).

Response

Thank you for appreciating the effort with this. It took us a very long time to gather this dataset from our own sampling in the field and that of all collaborators in the manuscript. We believe this manuscript and certainly this dataset is helpful to continue the discussion on plant functional traits but that it is also useful to test other hypotheses and answer new questions.

2. regarding different approaches to vegetation mapping. I see that my comments on neural network were not considered in the review. I think that given the prominence of AI in current research (and in particular deep learning) they deserve a tiny spot in your related works. There are now major works that are based on such technology (see eg. <https://www.nature.com/articles/s41559-023-02206-6.pdf> , <https://www.nature.com/articles/s41598-021-95616-0> , <https://www.nature.com/articles/s41477-021-01001-0>). It is ok that the authors don't want to compare (I would love to give it a try myself :)), but recent literature should still be acknowledged.

Response

We agree with you and I apologise for missing including this in our previous discussion. We have included a paragraph on this in the discussion section, see text below. Moreover, a co-author is working on understanding the role of SAR and LiDAR data on the prediction accuracy of these traits using FCNN vs RandomForest. We hope it's more comparative analysis will shed more light on the differences in predictions these other covariates and modelling approaches may produce.

Our new text is:

'Our capacity to use artificial intelligence to map plant functional traits by means of Deep Learning models applied to field trait ⁶⁵ data or even photographs ⁶⁶ is quickly developing. These models can process vast amounts of remote sensing data to identify and classify diverse biodiversity metrics ⁶⁷, and particularly convolutional neural networks, have been integrated with spectral data to map plant traits using field data ⁶⁸ and recently also citizen science approaches ⁶⁹. There have been recent developments of new satellites with hyperspectral and high spatial resolution capabilities and on the availability of large amount of tree censuses and trait data across the tropics, which opens new venues for the coming years for testing the capabilities of AI beyond machine learning approaches for mapping and monitoring, almost in real time, plant functional traits across the tropics. However, to obtain robust and reliable indicators of plant functional diversity and biodiversity levels across ecosystems AI models should complement and not replace traditional ecological methods - especially the direct field sampling and botanical identification of individual trees by experts.'

3. regarding spatial testing. Same for the spatial testing, there was no real response to my comments in the updated manuscript. I understand that random sampling gives better numbers, but for a work centered on large scale mapping, hence providing a map OUTSIDE of the plots area, random selection of pixel is not acceptable. Reading Table 3 in the supplementary, I see that the performances drop of 0.02-0.2 in terms of R², with an average of 0.1. I think the drops are acceptable in terms of performance, especially given the global scope of the paper, but I really believe these are the results that should

be reported in the main paper, and also used for all the maps and analysis. I understand that in Aguirre-Gutierrez et al. (2021) all the study was based on plot data, but here the objective is to evaluate a pantropical map that span across the plot data itself. The spatial testing results are the performance to be expected in the areas not in the plots and in this paper that is the majority.

Response

Based on your comment we have modified our approach and present the results of the spatial cross-validation approach, as also suggested by Ploton et al. 2020 Nature Communications. With this approach we account for spatial autocorrelation and give a more realistic account of the accuracy of our predictions, which overall continue being in the same range as before. We use the approach with all data and also focusing on predictions to Africa and Asia. As expected, given the lower sample sizes in these two continents the prediction accuracy drops. We therefore also suggest areas in these continents where we have more needs of new samples to improve any predictions made for plant functional traits (Supplementary Figure 15). We also show directly in the text the RMSE and include in the Supplementary Table 3 further results for the MAE and MSE. Therefore, we do not give an average of prediction accuracy anymore but the actual values obtained.

4. Going further down this reasoning, the plots geographical distribution is very skewed to the Americas, which the authors discussed in the paper at line 94. Looking at plots in Fig 1A and 1B, there are still quite some environmental conditions not covered by the dataset, so I am still wondering if one should not try a more thorough analysis of the results using spatial splitting. For example, I would suggest to run some experiments where plots from the Americas are used for training and those from Asia and Africa are used for testing. This way we will see if there is a systematic bias in the predictions (by comparing the performance of these models to the "globally trained" one).

Response

We acknowledge that the data is biased towards the Americas but also that our representation of environmental space across the tropics is good. However, given the data availability constraints for Africa and Asia, we make the assumption that trait-environment relationships are maintained, an assumption that is rooted in trait theory (Enquist et al. 2015). However, it may be that such trait-environment relationships are not maintained totally across the tropics, which is a hot research topic now. Acknowledging this, we have included in our results the text below and have also included the results of the spatial cross-validation approach for Africa and Asia (Supplementary Table 3) as suggested. The action points based on our results is to do more field sampling across Africa but especially in Asia where much less data is available and which would allow us to improve our models. It is important to highlight that we have gathered the most comprehensive dataset on vegetation plots and plant functional traits we know exists and we are committed to improving such dataset and our spatial predictions as more data becomes available. We also include a note about this in our conclusions:

Here the new text:

'As expected, lower explanatory values were found when testing the models with the plots from Africa or Asia separately for which less data were available (Supplementary Table S3).'

We argue that the lower accuracy for Africa and Asia in principle due to the limited capacity of machine learning methods, such as Random Forest, to predict well beyond the data used for model training but may also be the result of having relatively few data points in these areas for model testing in comparison to the Americas.

Following your comment we also updated the following text in our manuscript:

'...our maps represent data-informed spatial hypotheses that assist in identification of priority areas for further field collection, especially across tropical forests in Africa and Asia where less data is available. The ultimate accuracy of the plant functional trait predictions depends on the sample coverage, the accuracy of the field measurements, and the quality of the pantropical covariates used to spatially extrapolate our models. Undoubtedly, predictions will improve as new environmental datasets become available and as vegetation census and trait data expand further over space and time.'

Enquist, Brian J., et al. "Scaling from traits to ecosystems: developing a general trait driver theory via integrating trait-based and metabolic scaling theories." *Advances in ecological research*. Vol. 52. Academic Press, 2015. 249-318.

5. regarding the maps. I really like the maps and I think they are the added value of the paper (as stated in my previous review). And this is why I think the authors should make that one the main points of their discussion! What I mean is that the authors create these very high resolved maps of traits (10m, globally, it's massive), but then discuss mostly at regional or national scale. In the paper, we only see pan-tropical maps (e.g. fig 2 or in the supplementary), where one cannot really appreciate any of the details of specific regions. I am wondering if the same conclusions couldn't be reached with the plot data alone in many cases. What I am trying to say is that I would like to see cases where the patterns predicted allow some conclusions that we couldn't do with the plots data alone. I understand that some of the conclusions in the paper couldn't be done without those maps simply because there is no plot data, but having the discussion without visual examples hinders the appreciation I can have (again, as a remote sensing person). Maybe a figure with some examples would be enough to stress what is the actual plusvalue of these maps and the authors could provide a link to a map on Google Earth Engine where readers can zoom in and see the specific patterns.

Response

We understand your point here and have improved Figure 4 with your comment in mind. Because the single trait maps are too many to zoom in at a region, we have made clearer the added value of these products by giving a more detailed example of some regions in the Americas, Africa and Asia where we present the distribution of trait syndromes (Figure 4). Hence, based on your comment now we manage to show in fine detail how these traits syndromes are distributed at finer scale, which was not possible by only having the pantropical maps without a zoom in figure.

Moreover, following your advice we will provide, as an Earth Engine app, the trait maps and maps of syndromes online so that interested readers can work with them interactively. We have included new text in our manuscript to describe this as follows:

‘We make available our trait mapped predictions across the tropics as an online resource where more detail can be obtained across the tropical region (Link here).’

6. regrading the PCA. I don't understand why the black outlines in Figure 4A and 4B are different. I thought they were outlining the data cloud of the whole dataset (otherwise those in the three subplots of 4B should also be different from each other). Can you explain? Also, I think subplot C should report more than the 2 first PC only. There is still ~40% of information in the subsequent PCs and I am sure there are different and interesting patterns to be found there. I know it doesn't help for the 2D plots in panels 4A and 4B, but For subpanel C I would go a bit deeper.

Response

We agree, there was a plotting discrepancy for Figure 4A and the full environmental space was not correctly plotted. We have fixed this issue and Figures 4A and 4B now match. About your second point, we also agree and have included PC3 in Figure 4C. This PC3 explains 13% of the variance in the distribution of trait syndromes and brings about more information about leaf water content. We have also included zoom insets that show more local details of the predictions of trait syndromes, also focusing on your previous comments. This allows the reader to get more insights of how local areas look regarding plant trait syndromes.

7. Regarding the uncertainty maps. I see that they are reported in terms of standard errors, to maintain their original units. This is nice, I was wondering if it would be possible to add a relative version (e.g. reported in number of standard deviations for all maps). This way, one could average them and provide a "global" score of uncertainty that could be used to search for the most interesting places to search (as stated at line 206). There are works going in this direction (e.g. active learning: <https://ieeexplore.ieee.org/stamp/stamp.jsp?arnumber=6185660>) and such a map could be of great interest to drive future definition of measurement plots. Now one would have to use and weigh in... 12 maps. Alternatively, one could count on how many maps a certain uncertainty threshold is surpassed and use those as "interesting" areas.

Response

This is a great idea for having a general map of sampling needs. We have included our new approach, based on the data we have available, to obtain something similar to what you suggest. Our new text in the methods section and figure are as follows:

‘From these SE mapped predictions, we also calculated a final map of new field sampling needs by standardising each trait SE mapped prediction from 0 to 1 and obtaining an average value of the sum of those standardised SE maps. From this final field sampling needs map, we calculated the areas belonging to the lowest, middle and highest 33 percentiles and classified these as ‘Low’, ‘Intermediate’ and ‘High’ respectively. This final map could aid in generating field sampling priorities for the traits used in this study.’

The new Figure is in Supplementary Figure 15 and is show here:

Supplementary Figure. Field sampling needs based on the standardised scores of prediction error for trait predictions across tropical forests.

Referee #4

Narrative:

The authors put a lot of emphasis on the fact that the new feature maps can be useful for models. I do not argue against this as I fully agree. However, I feel that there is little discussion of this. I think there should be more emphasis on this aspect (and perhaps reference to papers showing that averaging parameters in large grid cells is not ideal, while having this description of functional diversity and trait variability is important).

Response

We agree and have included in the main text a paragraph that discusses the role of plant traits and in general of plant functional composition on models, specifically on DGVM and SDMs which are commonly used to map feedbacks in the biosphere and the distribution of species and important biodiversity variables.

Our new text is as follows:

‘Understanding the tree trait composition and functional diversity across the tropics is of pivotal importance for global biodiversity and ecosystems modelling and conservation efforts^{60, 61}. While dynamic global vegetation models (DGVM) and species distribution models (SDMs) help to assess impacts of a changing climate, DGVMs often rely on broad plant functional types and SDMs commonly overlook functional trait composition and diversity (but see⁶²). By incorporating trait-based mechanisms and functional trait diversity, models can better capture the variability in plant responses, potentially making more realistic predictions related to carbon cycling⁶³, vegetation distribution⁶⁴, and ecosystem composition and resilience⁴⁷. DGVMs and SDMs could include plant traits and plant functional diversity estimates to advance our understanding on ecosystem functioning and responses to global environmental change.’

Also, and this is more of a suggestion on how I would approach the discussion, I would put more emphasis and discussion on where diversity hotspots are located, if there are differences between protected and unprotected areas for example, and the conservation needs of these areas, rather than on the modelling part.

Response

Following your comment, we have restructured our manuscript, especially in the results and discussion sections. We have updated our text to include further details and discussion on the main patterns of the distribution of trait syndromes and functional

diversity. We especially focus on the importance of trait syndromes and functional diversity as indicators of forest stability and resilience to global environmental change and on the potential of our results to understand such resilience. We also focus on the relevant differences found in the distribution of traits across wet and dry tropical forests as this could have implications for forest resilience as shown by other recent studies cited in this section. Furthermore, as suggested, we have also decreased the amount of detail given to the single traits and the modelling, which has given us the opportunity to increase our discussion as mentioned above. We had to limit the amount of new text in the manuscript given editorial constraints on word count.

Line 92-96. I do not fully agree with the statement “While our vegetation plots are more abundant in the tropical forests of the Americas and it could be thought they mostly represent the environmental conditions in this region than in Africa and Asia, our principal component analysis (Fig. 1B and 1C) shows that with our pantropical sampling sites we cover the most prominent environmental conditions found across tropical forests in the world.”

The PCA analysis in Figure 1 B and C shows that there are large areas of the data space that are not covered by the sampling sites. It is true that the areas of higher density are covered, but these are likely to be the Amazon. In order to prove what the authors claim, I propose to show on the biplots the points that are located in Africa and Southeast Asia.

Response

Thank you for your suggestion. We have made the additions suggested and improved in this way our Figure 1. We also have modified the text to better acknowledge that there is more data in the Americas than in Africa and Asia. Please see our new Figure 1 and the new text that now reads:

‘Our vegetation plots are more abundant in the tropical forests of the Americas and it could be thought they mostly represent the environmental conditions in this region than in Africa and Asia. Our principal component analysis (Fig. 1B and 1C) shows that although our sampling sites do not cover all environmental space available across the tropics, especially those climates that are less common in the tropics (dark purple zone in Fig. 1B and 1C), we fundamentally cover the most prominent environmental conditions found across tropical forests.’

The geolocation error of Sentinel 2 is about 10 m according to Gascon et al. (2017). Therefore, the selection of only 1 pixel is not ideal. The authors should acknowledge this aspect and they can test the use of a 3x3 window for the training, or include multiple traits and average within a larger windows. Here the reference.

Gascon, F.; Bouzinac, C.; Thépaut, O.; Jung, M.; Francesconi, B.; Louis, J.; Lonjou, V.; Lafrance, B.; Massera, S.; Gaudel-Vacaresse, A.; et al. Copernicus Sentinel-2A Calibration and Products Validation Status. Remote Sens. 2017, 9, 584. <https://doi.org/10.3390/rs9060584>

Response

Thank you for your comment, this is certainly an important topic given the focus of our manuscript.

We believe there is a misunderstanding on the results of Gascon et al., 2017. In Table 13 of Gascon, they report a mean geolocation error of only 4.5 m for L1C and later state "After correction of the geometric model, the residuals errors were less than 0.2 pixels."

Both Sentinel-2 L1C and L2A products switched to an even further improved geolocation in August 2021.

https://sentinels.copernicus.eu/en/web/sentinel/missions/sentinel-2/news/-/asset_publisher/Ac0d/content/worldwide-extension-of-the-geometric-refinement-processing-on-sentinel-2-products-on-23-august-2021

The manuscript from Gascon et al. is from 2017 when the geometric calibration of the Sentinel-2 data was not yet implemented with the Global Reference Image (GRI). Clerc et al (2021) have described the GRI and the co-registration algorithm (geometric refinement) in a newer manuscript published in 2021 (see reference below). In this paper the authors show that to validate the geometric performance of Sentinel-2 they make use of a grid with about 4000 Ground Control Points (GCPs). Then they measure the shift between the image and the GCPs to assess the geometric performance of the products. Hence, we suggest that if the geometric corrections, especially that with the GRI, would have not been done then a selection of one pixel would not be ideal. However, given the improvements done to the geometric error using the GRI we are confident the selection of one pixel as the study unit is a good choice.

Clerc, Sébastien, et al. "Copernicus SENTINEL-2 geometric calibration status." *2021 IEEE International Geoscience and Remote Sensing Symposium IGARSS*. IEEE, 2021.
<https://ieeexplore.ieee.org/document/9555090/figures#figures>

<https://sentinels.copernicus.eu/web/sentinel/-/forthcoming-deployment-of-the-copernicus-sentinel-2-products-geometric-refinement>

We do not think using a 3x3 pixel window would be ideal as this per se would introduce a new error and noise into the model, making our predictions less reliable. Relating the weighted canopy functional traits to adjacent neighbourhood pixels would muddle the underlying signal between the spectral signature of the focal pixel, therefore almost certainly leading to a larger error term.

We are not clear as to what is meant by "include multiple traits and average within a larger windows". The analysis consisted of separate models for each canopy trait and mixing them would encumber the separate accuracy evaluation of the different models. A fundamental component of this analysis was to determine which canopy traits can be modelled from the Sentinel-2 multispectral and other environmental data.

Line 128-136 - These statistics are fundamental to understanding how reliable the model is. It is unclear what exactly these statistics are (mean R2 across different cross-

validation strategies?). In the methods the authors state: "The accuracy of the predictions was quantified by the explained variance using the R²". Is the R² in cross-validation at all? This should be clarified, and I suggest using the cross-validation statistic to show the predictive accuracy of the model (see also comments below on the unbalanced dataset). Also, at least a mean error (MAE) and a relative error (rMAE) should be included. The R² does not provide any insight into the error and does not help to understand how statistically sound the result is. The authors calculate the OOB RMSE, but this is never reported. Nevertheless, the error statistics should be included in the cross-validation. Moreover, given the large difference between the sample size in the Amazon and the sample size in the African and Asian continents, it would be important to design a cross-validation that allows understanding the confidence of the model in extrapolation in Africa. For example, the authors should test a training of the model in the Amazon and in Asia and predict the points in Africa and the same for Asia. Otherwise, the cross-validation, as currently designed, may be overoptimistic and suffer from the fact that the samples are so spatially unbalanced. Another option is to perform a balanced cross-validation. This can be done by maintaining the share of data from Amazon, Africa and Asia, in the training dataset, train the model and predict on the unseen data points, but then test the performance and present them by continent.

Response

We have taken into account your suggestion and have updated the text and the statistics shown. In our original manuscript we included the R², RMSE, MSE and MAE in the supplementary material (Supplementary Table 3), perhaps this was not clear in this section. We have updated the text as follows:

'We tested for the prediction accuracy and uncertainty in trait predictions while accounting for potential spatial autocorrelation using a plot level spatial block leave-one-out cross-validation³⁵ (Supplementary Table 3).

'Models for leaf chemistry and wood density displayed higher accuracy (mean R² = 0.66 and 0.48, respectively) than those for leaf morphology traits (mean R² = 0.25; Supplementary Table 3). Among these, leaf nitrogen (mean R² = 0.53/Root Mean Squared Error = 0.29), phosphorus (0.50/0.02) and calcium (0.64/0.22) concentrations had the highest prediction accuracy followed by leaf carbon (0.40/1.42) and potassium (0.46/0.17). Models for SLA (0.32/19.95), leaf dry (0.32/0.58) and fresh mass (0.31/2.24) demonstrated moderate accuracy scores. In contrast, leaf magnesium concentration (0.27/0.06), leaf area (0.22/66.15), leaf water content (0.18/3.92), and leaf thickness 0.17/0.79) had lower accuracy. As expected, lower explanatory values were found when testing the models with the plots from Africa or Asia separately as less data were available (Supplementary Table S3).'

We argue that the lower accuracy for Africa and Asia in principle due to the limited capacity of machine learning methods, such as Random Forest, to predict well beyond the data used for model training but may also be the result of having relatively few data points in these areas for model testing in comparison to the Americas.

The maps are made independently. For example, maps of N and P are produced without any constraint on the stoichiometric ratios that we know exist in the leaf sample. In my

opinion, another aspect that should be evaluated to prove the robustness of the approach is to calculate maps of N/P and C/N ratios and evaluate if these ratios are in the ballpark of what is measured in the field.

Response

This is a great suggestion. We have carried out the analysis requested and show that the values for both N:P and C:N are in ballpark of what is expected based on field data. Please see below the resulting maps and also the references to the values expected:

Ratio N:P results

Statistics: min 14, median 20, max 26;

Ratio C:N results

Statistics: min 18, median 22, max 32;

References:

N:P overall values between 10 and 30.

Townsend, Alan R., et al. "Controls over foliar N: P ratios in tropical rain forests." *Ecology* 88.1 (2007): 107-118.

https://esajournals.onlinelibrary.wiley.com/doi/full/10.1890/0012-9658%282007%2988%5B107%3ACOFNRI%5D2.0.CO%3B2?saml_referrer

Vallicrosa, Helena, et al. "Global distribution and drivers of forest biome foliar nitrogen to phosphorus ratios (N: P)." *Global Ecology and Biogeography* 31.5 (2022): 861-871.

<https://onlinelibrary.wiley.com/doi/full/10.1111/geb.13457>

Specht, Ray L., and Alison Specht. "The ratio of foliar nitrogen to foliar phosphorus: a determinant of leaf attributes and height in life-forms of subtropical and tropical plant communities." *Australian Journal of Botany* 58.7 (2010): 527-538.

<https://www.publish.csiro.au/bt/Fulltext/BT07110>

Manu, Raphael, et al. "Response of tropical forest productivity to seasonal drought mediated by potassium and phosphorus availability." *Nature Geoscience* (2024): 1-8.

<https://www.nature.com/articles/s41561-024-01448-8>

C:N and N:P

López-Camacho, René. "Litterfall and nutrient transfer dynamics in a successional gradient of tropical dry forest in Colombia." *Revista de Biología Tropical* 71.1 (2023).

https://www.scielo.sa.cr/scielo.php?script=sci_arttext&pid=S0034-77442023000100046

C:N

McGroddy, Megan E., Tanguy Daufresne, and Lars O. Hedin. "Scaling of C: N: P stoichiometry in forests worldwide: Implications of terrestrial redfield-type ratios." *Ecology* 85.9 (2004): 2390-2401.

<https://esajournals.onlinelibrary.wiley.com/doi/10.1890/03-0351>

Figure 1. Predicted N:P and C:N ratios based on the prediction of single traits across tropical forests. The values are well in range from what is expected from our own trait sampling collections across the sampling plots in the tropics and also in agreement with those reported in the references shown above in the response to your comment.

The authors have analysed the uncertainty of the prediction, which is a great addition. However, I would suggest that the authors plot the trait maps (left column) and the trait uncertainty maps (right column). For each trait, I would keep the same colour scale for the trait map and the uncertainty map. This way the reader can see how large the uncertainty is compared to the trait values.

Response

We have reconfigured the figures as you suggest. However, given the size of the maps we have included these figures in the SI and not in the main text. Hence, we now include the predicted traits and their prediction error as maps one in the top panel and other in the bottom panel. We also tested if having the same colour legend would be appropriate, however, given the different ranges in values between the mean prediction and the prediction error the colour range tended to shrink substantially in the prediction error maps making it virtually impossible to see much difference across regions for those maps. Hence, we decided to use the native value ranges for the colour legend in each map. The new figures are in the Supplementary Information Figs. 2-14

Minor comments

Referee #2

Below are a couple of minor questions.

In the functional richness analysis, to what extent are the observed differences across continents influenced by the differences in sampling efforts? I don't disagree with the authors' discussion of the differences in functional syndromes across continents, e.g.,

leaf phosphorus and other nutrients (lines 305-344). But the differences in sampling efforts are quite apparent (Figure 1). It would be beneficial if the authors could conduct a similar analysis using only the field data.

Response

We agree about the differences in sampling effort and have tried our best to bring more data into our analyses for all regions, which we have managed to do in comparison to the first version where we only had 74 plots and now 1814. Despite our efforts we did not manage to obtain as much data for Africa and Asia as we have done for the Americas and hence, we recommend along our manuscript, and certainly in the conclusions, to improve sampling across these regions. Our text in the conclusion section is:

'...our maps represent data-informed spatial hypotheses that assist in identification of priority areas for further field data collection, especially across tropical forests in Africa and Asia where less data is available. The ultimate accuracy of the plant functional trait predictions depends on the sample coverage, the accuracy of the field measurements, and the quality of the pantropical covariates used to spatially extrapolate our models. Undoubtedly, predictions will improve as new environmental data sets become available and as vegetation census and trait data expand further over space and time.'

In principle, the representativeness of our sampling effort for across the tropical forest's environmental conditions (as shown in Figure 1 B and C) should allow us to build well data informed models of the expected trait-environment relationships found across the study area. However, given the differences in sampling effort, which you are right in pointing and questioning, we do not think the analysis shown in Figure 4 should be done with the raw unbalanced data as indeed Africa and Asia would have much less sampled environmental conditions than the Americas. However, using the spatial predictions (maps) to carry out the analysis from Figure 4 avoids the issue of differences in sampling at this stage as we are now comparing the distribution and arrangement of plant syndromes across the predicted areas (Americas, Africa and Asia).

Figure 1C, many field plots are characterized by positive scores along PC3. They appear to be outliers relative to the environmental space. Comments?

Response

We thank you for your comment and great observation. Based on your comment we have included the environmental variables in the PCA plots to improve the understanding of the figure and of the distribution of environmental conditions and sampling plots. The areas you mention are representing mostly locations with lower soil moisture and lower maximum temperatures, characteristics found across the higher elevation plots in the study area. Such locations are therefore not outliers and are of great importance for the analyses and results of the study.

Referee #3

Very minor things:

- sometimes the abbreviations are in square parenthesis, sometimes in round parenthesis, please normalise

- can the uncertainty maps made as big as the prediction maps? I really appreciated that the prediction maps were given space in the supplementary material and I don't understand why the uncertainty maps are relegated into minuscule plots.

Response

We are now using only round parentheses and we have made the uncertainty maps into the same size as the prediction maps and both of them are nested in the same figure (Supplementary Figures 2-14) for ease of comparison.

Response to remaining comments from reviewers for manuscript 2022-06-08600B:

“Canopy functional trait variation across Earth’s tropical forests”

Response to comments from Reviewer 3:

1. Since it is mentioned in the intro that there is a need for tools with high temporal resolution, I would appreciate a short paragraph in the discussion mentioning what would be needed to achieve that. I understand this is beyond the scope of this paper, but it would be nice for people willing to expand on this paper to know what would be necessary to turn this into a monitoring tool. Also, an estimate of the data and compute needs would be great if possible.

Response: To accommodate the recommendation from the reviewer, we have updated our discussion, on the penultimate paragraph as follows:

‘There is a need for tools that can generate predictions of biodiversity at high temporal resolution and our approach represents a way forward in this direction. Going forward, there is the potential to track plant functional diversity across time, e.g. on a yearly basis, using satellite remote sensing data such as that from the Sentinel-2 satellites. Such an application would certainly require major efforts on field ecological data collection, availability of new satellite data, modelling algorithms, computing power and storage capabilities. All of this can be achieved by strong and fair collaborations between field researchers, universities and other public and private research organisation.’

2. "AI beyond machine learning approaches": for me, "AI" is more of a fashionable term, nowadays corresponding to the usage of ML models (and the most often involving deep learning). Maybe you want to rephrase in to something like "testing the capabilities of large machine learning models, possibly involving deep learning, and enabling the capabilities of using data from multiple sources. A reference for that could be the SAtBird paper (<https://satbird.github.io>).

Response: We have rephrased the sentence accordingly

3. "Almost in real time": well, I would be a bit careful in overpromising that. LARge scale DL model come with strong computational needs and also will require a lot of data downloads and preprocessing (see also my comment in 1. above). I would probably not mention this and simply state that we can have a temporally-aware, multi-sensor series of interesting models based on recent powerful advances in AI.

Response: We have deleted that phrase based on the comment from the reviewer.

4. I love your last point about not forgetting ecological models! I think this is where the field of hybrid DL for ecology should go! There are a couple of examples around (for example taxonomy-aware models <https://www.sciencedirect.com/science/article/pii/S0924271621002641> or models for structured biological knowledge <https://imageomics.github.io/bioclip/>). Maybe worth adding those to the discussion?

Response: We had this in the discussion and based on the limit on the number of references sent by the editorial office we decided not to include the references suggested.

5. line 370: please add a comma after "traits", I think it would ease reading

Response: Done

6. line 454: "capabilities against overfitting and variable inputs"  "capabilities against overfitting and for its flexibility with respect to variations in the type and number of variable inputs"

Response: Done

7. line 462: "model"  "models" (or "model for each trait")

Response: Done

line 474: remove "at"

Response: Done

8. lines 480-484: I would put those after the paragraph ending at line 471, since they pertain to the traits prediction, not the uncertainty.

Response: Done

9. Supplementary table 7. I appreciate the authors making snippets of code available, but why not giving a link to a github with some sample code? This would have the advantage of potential updates and avoid obsolescence of your code.

Response: We now provide the code to generate main outputs of the manuscript in R.

10. Line 148: why leaving the "Link here"? I would have loved to check the online resource.

Response: We are now giving a link to a EarthEngine app where the maps can be visualised.